



# Implementation of the Blade Element Momentum Model on a Polar Grid and its Aeroelastic Load Impact

Helge Aagaard Madsen[1], Torben Juul Larsen[2], Georg Raimund Pirrung[1], Ang Li[1], and Frederik Zahle[1]

[1]Technical University of Denmark, Wind Energy Divison, Building 118, P.O. Box 49, 4000 Roskilde, Denmark
[2]Vestas Wind System A/S, Hedeager 42, 8200 Aarhus, Denmark

**Correspondence:** Helge Aagaard Madsen (hama@dtu.dk)

**Abstract.** We show that the upscaling of wind turbines from rotor diameters of 15-20 m to presently large rotors of 150-200 m has changed the requirements for the aerodynamic Blade Element Momentum (BEM) models in the aeroelastic codes. This is because the typical scales in the inflow turbulence are now comparable with the rotor diameter of the large turbines. Therefore the spectrum of the incoming turbulence relative to the rotating blade has increased energy content on 1P, 2P, …, nP and the

annular mean induction approach in a classical BEM implementation might no longer be a good approximation for large rotors. We present a complete BEM implementation on a polar grid that models the induction response to the considerable 1P, 2P, …, nP inflow variations, including models for yawed inflow, dynamic inflow and radial induction. At each time step in an aeroelastic simulation the induction derived from a local BEM approach is updated at all the stationary grid points covering the swept area so the model can be characterized as an engineering actuator disc (AD) solution. The induction at each grid

point varies slowly in time due to the dynamic inflow filter but the rotating blade now samples the induction field; as a result the induction seen from the blade is highly unsteady and has a spectrum with distinct 1P, 2P, …, nP peaks. The load impact mechanism from this unsteady induction is analyzed and it is found that the load impact strongly depends on the turbine design and operating conditions. For operation at low to medium thrust coefficients (conventional turbines at above rated wind speed or low induction turbines in the whole operating range) it is found that the grid BEM gives typically 8-10 % lower 1 Hz fatigue

loads than the classical annular mean BEM approach. At high thrust coefficients the grid BEM can give slightly increased fatigue loads. In the paper the implementation of the grid based BEM is described in detail and finally several validation cases are presented.

## 1 Nomenclature

$a$     axial induction factor

$a'$     tangential induction factor

$A$     rotor area

$c$     chord

$C_d$     sectional drag coefficient



$C_l$     sectional lift coefficient

$C_Q$     rotor torque coefficient

$C_T$     rotor thrust coefficient

$C_x$     projection of $C_l$ and $C_d$ tangential to the rotor plane

$C_y$     projection of $C_l$ and $C_d$ perpendicular to the rotor plane

$dT$     rotor thrust on a ring element

$dQ$     rotor torque on a ring element

$F$     Prandtl tip correction factor

$k_1, k_2, k_3$     polynomial coefficients in $C_T(a)$ curve fit

$k_a$ reduction factor of induction for yawed flow

$k_x, k_y$     parameters for azimuth variation of induction in yaw model

$N_B$     the number of blades

$r$     radial position

$R$     rotor radius

$T$ rotor thrust

$\mathbf{T}_{G \rightarrow S}$     transformation matrix from global to sectional coordinates

$\boldsymbol{u}_{i,y}$     induced velocity vector in axial direction

$u$     non-dimensional axial velocity (velocity divided with $U_0$)

$U_0$     free stream velocity

$\bar{U}_0$     free stream velocity vector

$\bar{U}_{0,l}$     local free stream velocity vector

$\boldsymbol{U}_{grid}^{G}$     free wind speed vector at grid point (global coordinates)

$\boldsymbol{U}_{i,grid}^{G}$     induced wind speed vector at grid point (global coordinates)

$\boldsymbol{U}_{grid}^{S}$     resultant relative flow speed at grid point (section coordinates)

$\dot{\boldsymbol{x}}_{blade}^{G}$     blade velocity vector at grid point (global coordinates)


**Greek letters**

$\rho$     air density

$\varphi$     angle in Prandtl tip correction factor

$\Phi$     yaw angle

$\chi$     wake skew angle

$\Psi$     azimuth angle





## 2 Introduction

The Blade Element Momentum (BEM) model (Glauert, 1935) is used extensively within the wind energy research community
as well as by the wind turbine industry for simulating the aerodynamic rotor characteristics such as blade aerodynamic loads,
rotor power and rotor thrust. For rotor design the computations are commonly carried out for uniform, constant inflow over the
rotor disc. However, the BEM model is also the aerodynamic engine in most aeroelastic models used today (FLEX5 (Flex4)
(Øye, 1996), FAST (Jonkman et al., 2016), BLADED (Bossanyi, 2003), GAST (Riziotis and Voustinas, 1997), Cp-Lambda
(Botasso and Crooce, 2006-2013), FOCUS (WMC, 2019), HAWC2 (Larsen and Hansen, 2007)) by the industry for the detailed
aeroelastic simulations that are the basis for the certification of wind turbines (Hansen et al., 2015; IEC, 2005). This comprises
a significant amount of simulations at normal operating conditions with turbulent inflow but also at fault modes of the turbines
such as a large yaw error. It further includes extreme inflow conditions such as strong shear, gusts and more recently also wake
situations, where the wake is modeled as a combination of a reduced, meandering wind speed deficit in the wake region and
added wake turbulence (Larsen et al., 2008), (Madsen et al., 2010c).
When describing an aeroelastic code, it is often just mentioned that BEM is the model for computing the aerodynamic forces
and that the model is further extended with sub-models for tip loss, yawed conditions, dynamic inflow and dynamic stall.
This is an incomplete description as implementation details such as the way the models are coupled together can influence the
computational results considerably. The most important aspect is how the BEM model is implemented to model the induction
response due to the unsteady and non-uniform loading over the rotor caused by the atmospheric turbulent inflow, wind shear
or control actions like pitch and flap control.

The purpose of the present article is to present in detail a complete unsteady BEM induction model for non-uniform inflow and
loading that can be readily implemented.

### 2.1 Upscaling has influenced the requirements for aerodynamic modeling

The non-uniform unsteady loading over the rotor disc due to the atmospheric inflow increases with rotor size. Thus the require-
ments to the BEM modeling capability have changed considerably from the 15-20 m diameter rotors in the nineteeneighties
to 100-200 m rotors today. This important effect of turbulence scales relative to rotor size was already described by de Vries
(1979) noticing the difference in impact on aerodynamic loads of turbulence scales above and below the rotor size. He also
very briefly presented how to use the BEM method in sheared inflow. His approach has some resemblance with the BEM
implementation that will be presented here.
To illustrate how the upscaling of rotors leads to a more non-uniform inflow and thus non-uniform loading of the rotor when
operating in turbulent inflow (no shear), we simulate two turbines with the aeroelastic code HAWC2 (Larsen and Hansen,
2007): the AVATAR rotor with a diameter of 205 m (Sieros et al., 2015) and a downscaled version of the AVATAR rotor with
a diameter of 51.4 m. Both turbines were simulated without tilt, a stiff structure and both operating at the same tip speed of
74.7 m s$^{-1}$ and in the same turbulent inflow with no shear. The turbulent inflow was generated with the Mann model (Mann,
1994) using a box with vertical and horizontal side lengths of 240 m and 5600 m, the latter corresponding to the travelling





length of the turbulence over the simulation time of 700 s and a mean wind speed of 8 m s$^{-1}$.

As the turbine blades rotate through the turbulent vortex structures, the spectrum of the free wind speed at the tip of the blades has energy concentrated on multiples of the rotational frequency 1P, Fig. 1. Since the size of the turbulent vortex structures is absolute (given a certain turbulence length scale) the distribution of energy upon the individual frequency multiples is different

for different turbine sizes. What can be noticed is that the small rotor has a significant amount of energy on the very low frequencies ($\ll$1P), whereas the larger turbine experiences a higher ratio of the total energy on 1P and multiples. In other words, the turbulent vortices no longer cover the full rotor for large turbines.

The increasingly non-uniform rotor loading with turbine size is also caused by inflow with shear. The largest modern turbines with the blade tips at top positions around heights of 300 m now span most of atmospheric boundary layer containing the main

part of the shear, Pena Diaz et al. (2009). This is in particular seen for stable flow situations.

Other challenging wind situations comprise nonstationary wind conditions containing trends, such as wind shear developing over time. For very large rotors these situations are important for the extreme load levels during operation. Thus they need attention in the modeling phase if turbine designers shall be able to counteract such events using either active or passive load alleviation techniques.

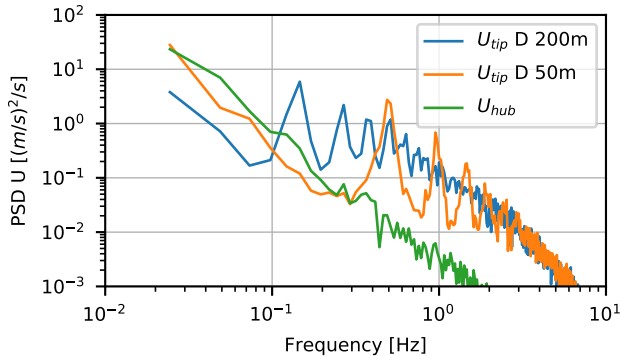

**Figure 1.** Rotational sampling of turbulence for different turbine sizes.

Besides the upscaling trend, turbine design has changed in the same time span of years which results in new requirements for the aerodynamic modeling in the aeroelastic codes. Pitch control is now the common power regulation method, therefore situations like pitch fault have to be simulated for certification. Such a situation with e.g. one blade pitch differing from the pitch of the other blades with e.g. 20° results in a non-uniform rotor loading and expected azimuthal variation of induction. The pitch control for power regulation has been extended to include cyclic pitch to alleviate 1P loads and now full individual

pitch is being implemented for even better load alleviation. An important question is thus how to handle individual pitch action in a BEM type modeling.



## 2.2 Research on the challenges in modeling sheared and turbulent inflow

The subject was part of the work in the EU funded UpWind project (2006-2011) with the main objective to study upscaling of turbines to 8-10 MW. The aerodynamic flow mechanisms at high shear in the inflow were investigated by simulating the sheared inflow on the 5 MW reference wind turbine (Jonkman et al., 2009) with a range of models from high fidelity CFD codes to vortex codes and to more BEM type codes (Madsen et al., 2012). One major finding was that all high fidelity codes and vortex codes showed that the induction does vary within an annular element for sheared inflow. Different BEM implementations to cope with this were discussed.

Similar work was continued in the EU funded AVATAR project (2013-2017) with focus on even bigger turbines (10+ MW) than in the Upwind project. A summary of the findings has been presented by (Schepers et al., 2018a). One major finding was that a comparison of aeroelastic simulations with a free vortex code and a BEM based aeroelastic code showed an overprediction of fatigue loads in the range of 15 % by the BEM based aeroelastic code. It is further mentioned and discussed that the results depend on the actual implementation of the BEM model.

## 2.3 The historical BEM development

The basic BEM formulation originates from Glauert and was developed for airplane propellers (Glauert, 1935). Glauert points out that the two major components are the "momentum theory" and the "blade element theory" which for many years were developed separately and e.g. the use of finite aspect blade data was considered in the blade element theories to fit experimental rotor data. However, the combination of the two theories finally led to the BEM approach where the induced velocities at the rotor disc are derived from the momentum theory and the blade sectional forces are found on the basis of infinite aspect ratio (2D) airfoil characteristics. In the present paper the focus is on the momentum part of the BEM approach although many uncertainties in rotor computations are linked to the blade element analysis such as 3D airfoil characteristics due to rotational effects.

When the momentum part of the BEM theory is used in aeroelastic simulations, the actual flow conditions violate most assumptions in the basic theory: 1) Turbulent and sheared inflow compared with the assumption of uniform, steady inflow; 2) Non-uniform load in contrast to the assumed uniform loading and 3) Skewed inflow in contrast to assumed axial inflow, just to mention the most important violations. To compensate for this, a number of sub-models are introduced like dynamic inflow and skewed wake models. However, there is no real consensus on how the different phenomena should be modeled and how the sub-models should interact. Therefore we often see considerable deviations for BEM simulations on complex inflow cases, (Hand et al., 2001; Schepers et al., 2018b).

Many researchers have over time contributed to the development of the BEM theory for wind turbines but only a few will be mentioned here. (Wilson et al., 1974) made an important contribution at an early stage to describe the theory. They also proposed a method based on a Taylor expansion to look into the effect of wind shear. Another important contribution at an early stage to the development of the BEM approach is made by de Vries (1979). He envisioned the challenges in implementing the BEM theory for turbulent inflow.



Later a comprehensive description of the BEM modeling is presented in the handbook of (Burton et al., 2011) with a detailed discussion of inflow models to handle dynamic and skewed loading as will be discussed later. Also the handbook of (Hansen, 2015) gives a fundamental introduction to the BEM modeling approach as well as the Doctoral thesis by (Sørensen, 2016).

## 2.4 The organization of the paper

In Sect. 3 we present a detailed description of the implementation of the grid BEM approach. However, first in that section we
give a short introduction to the origin of the CFD simulations of the actuator disc flow used heavily in developing and tuning the sub-model for yawed flow, the dynamic induction model and a sub-model for radial induction. The mechanism of induction in turbulent and sheared flow is explored in Sect. 4 and we present the load and power impact for two turbines for DLC 1.2 load cases. In Sect. 5 a selection of validation cases are presented followed by a concluding Sect. 6.

## 3 The grid BEM model implementation

The overall idea with the present BEM implementation is to model the rotor as an actuator disc (AD) that is updated at each time step in stationary grid points covering the rotor disc. In an aeroelastic simulation, the loading will normally be non-uniform and unsteady as discussed above. The input to the computation of the induced velocities is thus the distributed normal and tangential loading on the AD, and it will be shown in Sect. 3.4 how the loading of the individual rotor blades is distributed over the disc. The flow field could be computed with a CFD model of the AD (Madsen, 1999) but we present here an engineering solution
based on the BEM theory for the flow at the disc to reduce the computational efforts to a minimum. However, the close link between the engineering BEM-AD and the CFD-AD model means that we easily can tune sub-models in the BEM-AD model. We use below CFD-AD results for tuning the yaw and dynamic inflow model and for correction of the momentum model at high loading. Another example of such sub-model tuning is the correction for the influence of wake rotation and expansion (Sharpe, 2004) as presented by Madsen et al. (2010a). However, these sub-models are not incorporated in the BEM-AD model
presented here. Before going to the description of the BEM model, we will briefly introduce the origin of the AD CFD results used below.

### 3.1 The basis of the AD CFD results

The general purpose CFD code FIDAP, based on the finite element method, is used for the AD computations. It was one of the first commercially available CFD codes and has an unstructured mesh capability which reduces the requirements to the total
number of nodes.

In the past the code has been used for several studies of the flow through an actuator disc model. In a first set-up from 1996, the computations show good correlation with the momentum theory with 1/3 induction at the rotor disc and 2/3 in the far-field for a prescribed uniform loading corresponding to a thrust coefficient of 0.89 (Madsen, H. Aa., 1997). It should be mentioned that FIDAP has an option of using a discrete pressure formulation from element to element which was found important for AD
simulations of the pressure jump at the disc. Typically two cells with a total axial distance of 0.05R are used to model the disc





in axial direction.

Later in 1999, the AD model was coupled to the aeroelastic code HAWC (Thirstrup Petersen, 1996) so that the computation of the induction can be shifted between BEM and the CFD AD model (Madsen, 1999). Several yawed flow cases for a 100 kW turbine were investigated with that model set-up and a good correlation with experimental data was found, e.g. for the electrical power and flapwise moment (Madsen, 2000). A further comparison was made using the data set for 45° yaw error from the NREL Phase VI 10 m wind turbine tested in the NASA Ames 80 ft x 120 ft wind tunnel, (Hand et al., 2001). The computed angle of attack variation at a radial position of 83 % show good correlation with the measured inflow angle when corrected for the influence of upwash.

The CFD mesh and model from this set-up is used for the present simulations with a prescribed uniform loading on the disc, Fig. 2. The mesh extends 10 rotor diameters (D) in the z-direction which is the main flow direction for zero yaw, 12D in the x-direction and 4D in the y-direction. The inflow plane is 4D upstream the actuator disc and yawed flow is simulated by changing the inflow direction with an x-velocity component. In total about 25.000 nodes are used for the meshing.

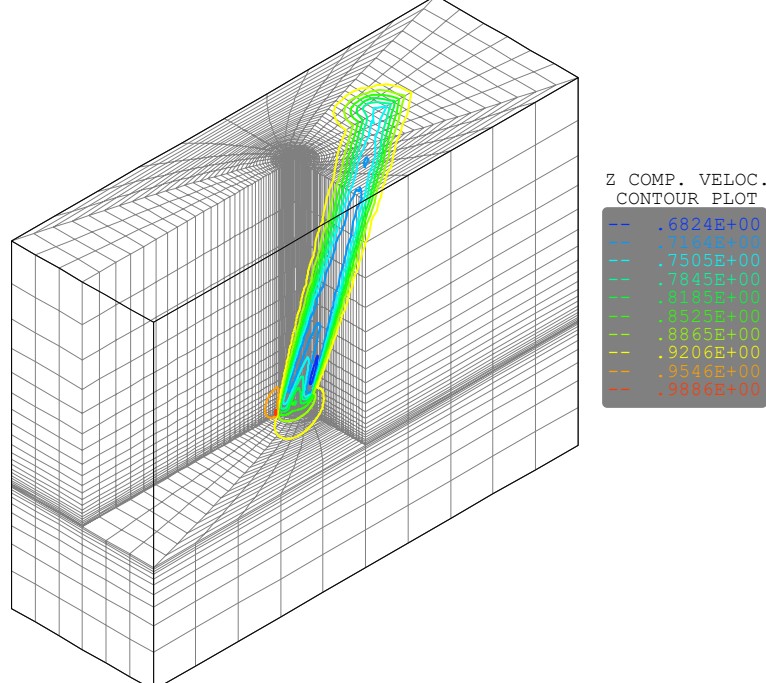

**Figure 2.** The CFD mesh used for the AD yaw computations. The velocity contours for computation of a 30° yawed case is shown on top of the mesh plot.

## 3.2 The basic BEM equations

The fundamental part of the BEM model (Glauert, 1935) is the relation between thrust on the rotor and the induced velocities. For a streamtube enclosing the AD a 1D momentum balance between axial forces on the turbine and the flow within a stream



tube is $T = \dot{m}\Delta U$. Following classical literature like Glauert (1935), Wilson et al. (1974) and de Vries (1979), this leads to the relationship between the thrust coefficient $C_T$ and the induction factor $a$:

$$C_T = 4a(1-a) \qquad (1)$$

where $a = \frac{U_0 - U_r}{U_0}$ and $C_T = \frac{T}{\frac{1}{2}\rho A U_0^2}$ with the rotor thrust $T$, the free stream velocity $U_0$, the air density $\rho$ and the rotor area $A$.

For thrust values causing higher induced velocities than $a = 0.5$, Eq. (1) breaks down since the flow velocity in the wake far downstream according to the momentum theory is $(1 - 2a)$ which in this cases is equal to or smaller than zero. This results in an infinite expansion of the flow behind the rotor and the flow can no longer be approximated by simple momentum theory. More complex flow models are needed such as CFD or an empirically based relation can be used.

For different reasons explained below we use a BEM implementation where the induction in the whole operational range from negative $C_T$ to a high positive $C_T$ is expressed through the following third-order polynomial shown in Fig. 3:

$$a = k_3 C_T^3 + k_2 C_T^2 + k_1 C_T \qquad (2)$$

where the coefficient $k_1 \ldots k_3$ are defined: $k_1 = 0.2460$, $k_2 = 0.0586$ and $k_3 = 0.0883$

For $C_T < 0.89$ the polynomial fits well the momentum equation. At high loading the curve was determined to fall between the Glauert empirical relation fitted to experimental results for a model rotor (see e.g. Burton et al. (2011)) and results from actuator disc simulations at high loading (Madsen, H. Aa., 1997).

One important reason for using a polynomial fit to Eq. (1) is that we find that it is a more robust and fast method to compute the induction instead of solving Eq. (1) using a non-linear Newton-Raphson iteration solver combined with an empirical relation at high loading. Another reason is that it makes it easily possible to modify this $C_T(a)$ relation in order to simulate e.g. a coned rotor as illustrated in Madsen et al. (2010a), using AD-CFD simulations for the coned rotor. In this case, the $C_T(a)$ polynomials will be a function of radial position.

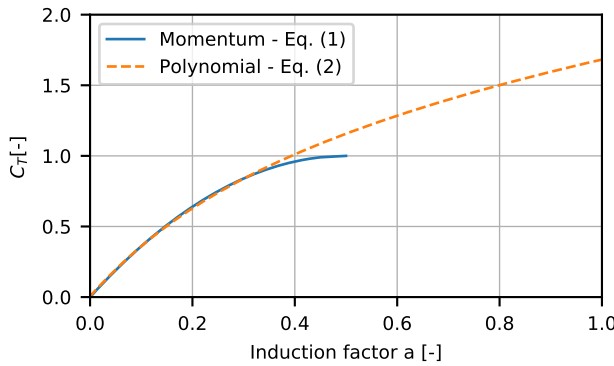

**Figure 3.** The approximation of the basic momentum relation between $C_T$ and $a$ with a polynomial extending into an empirical relation at high loading when $C_T$ is above 0.89.



A next step in implementing the BEM model is to couple the momentum theory to the blade element theory where the forces on a blade section are derived by means of two-dimensional airfoil characteristics. We apply Eq. (1) on a ring element of the rotor with the radial extension $dr$ as illustrated in Fig. 4 (left drawing):

$$C_T = \frac{dT}{\frac{1}{2}\rho A U_0^2 2\pi r dr} = \frac{U_{rel}^2 C_y c N_B}{U_0^2 2\pi r} \tag{3}$$

where $U_{rel}$ is the relative velocity to the blade section, $c$ is the blade chord, $N_B$ is the number of blades and $C_y$ is the projection of the lift coefficient $C_l$ and the drag coefficient $C_d$ on a line perpendicular to the rotor plane.

Besides the elemental thrust $dT$ on the ring element there is also a torque $dQ$ and we can define a torque coefficient $C_Q$ by:

$$C_Q = \frac{dQ}{\frac{1}{2}\rho A U_0^2 r 2\pi r dr} = \frac{U_{rel}^2 C_x c N_B}{U_0^2 2\pi r} \tag{4}$$

where $C_x$ is the projection of the lift coefficient $C_l$ and the drag coefficient $C_d$ on a line tangential to the rotor plane.

Applying the angular momentum equation across the disc we get:

$$dQ = \rho(2\pi r dr) r U_0 (1-a)(2ra'\omega)\, dr \tag{5}$$

Combining Eq. (4) and (5) we find:

$$a' = \frac{V_r^2\, C_x(\alpha)\, c\, N_B}{8\pi r^2 (1-a) U_0 \Omega} \tag{6}$$

where $a'$ is the tangential induction coefficient.

### 3.3 Tip correction

The relation between thrust and induced velocities, Eq. (1 and 2), is changed due to the presence of tip effects, caused by a finite number of blades. Within the wind turbine research community the tip correction method has for a long time been subject of numerous investigations and development. In a recent work, Sørensen (2016) presents a comprehensive review of studies on 230 the tip correction and contributes with full derivation of the commonly used Prandtl tip correction which however was further slightly modified by Glauert (1935) to be used in the BEM theory. The Prandtl tip correction factor $F$ as presented by Glauert (1935):

$$F = \frac{2}{\pi} \cos^{-1}\left( \exp\left( -\frac{N_B}{2} \frac{R-r}{r\sin\varphi} \right) \right) \tag{7}$$

We insert it into the the momentum equation (1) as:

$$\frac{C_T}{F} = 4a(1-a) \tag{8}$$

where $\frac{C_T}{F}$ has to be inserted instead of $C_T$ in Eq. (2). How to incorporate the tip correction factor is also discussed by Sørensen (2016), concluding that it can either be used to modify the circulation (loading) as done here or through a modification of the induced velocities.



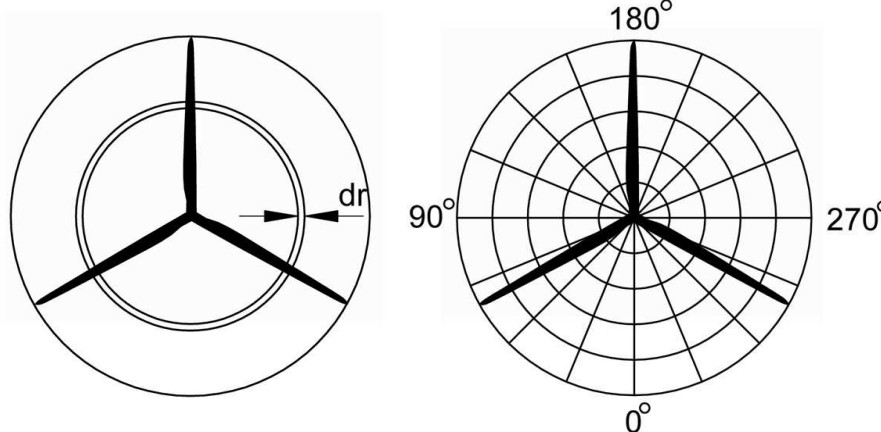

**Figure 4.** Illustration of the BEM approach. Left: Classic approach using an annular element to which the load is assumed constant over the element (mean value of blade forces). Right: New induction grid with annular elements and further subdivided azimuthally

### 3.4 Specific grid BEM implementation in HAWC2

Even though the BEM relationship is originally derived for a full rotor, it is generally implemented on an annular element form as proposed by Glauert (1935). In such an annular BEM implementation it is assumed that the loading and induction within each annular element are constant and that the annular elements are independent of each other. The $C_T$ coefficient now represents the average axial loading of the blades on an annular ring element.

In order to model azimuthal variations of induction due to azimuthal variations of blade loading as discussed above we propose
to expand the annular BEM approach. Dividing the annular elements into azimuthal sub-elements leads to a polar grid BEM approach, see the right part of Fig. 4. The induced velocity is found in each grid point using the $a(C_T)$ relationship in Eq. (2). For a uniform inflow and loading this leads to the exact same induction as the classic annular element approach, whereas differences are seen for non-uniform wind loading over the rotor. An important part of this azimuthal annular element approach is the definition of the local induction factor, where the local instantaneous induced velocity at a point in the grid is normalized
with the local free wind speed (the wind speed without rotor induction) at the exact same point.

$$a \equiv -\frac{\boldsymbol{u}_{i,y}}{|\boldsymbol{U}_{0,l}|} \tag{9}$$

As seen in Fig. 4, a question arises about how to find the local load in grid points that are not at the location of a blade. For the classic annular element the blade loads are averaged and the resulting blade load is assumed to be constant over the annular element. The solution for the azimuthally divided annular element (grid point) is to compute two different thrust coefficients
and torque coefficients. These coefficients use the pitch and velocities of the two neighboring blades combined with the local wind speed and induction at the grid point. The coefficients will be weighted by the azimuthal distance of the respective blades. For the corresponding radial position on these two blades, the transformation matrix from sectional to global coordinates is





rotated by the azimuthal distance between the blade and the grid point. This corresponds to virtually rotating the blade position to the position of the grid point. The blade velocities are rotated as well such that for example the velocity in the direction of

rotation at the blade location is applied as a velocity in the direction of rotation at the grid point. Then the flow at the grid point can be computed as the sum of the free wind speed $\boldsymbol{U}^G_{grid}$, the induced velocity $\boldsymbol{U}^G_{i,grid}$ and the rotated velocity of the blade section $\dot{\boldsymbol{x}}^G_b$; the latter has a negative sign because the flow will be experienced in the opposite direction of the blade movement.

$$\boldsymbol{U}^S_{grid,b} = \mathbf{T}_{G\rightarrow S}(\boldsymbol{U}^G_{grid} + \boldsymbol{U}^G_{i,grid} - \dot{\boldsymbol{x}}^G_b) \tag{10}$$

The superscripts $S$ and $G$ denote sectional and global coordinates, and the subscript $b = 1,2$ denotes the two closest blades.

The angle of attack $\alpha_b$ is computed by:

$$\alpha_b = \arctan 2 \frac{\boldsymbol{U}^S_{grid,b,y}}{-\boldsymbol{U}^S_{grid,b,x}} \tag{11}$$

and the relative velocity $U_r$ by:

$$U_{r,b} = \sqrt{\boldsymbol{U}^{S^2}_{grid,b,x} + \boldsymbol{U}^{S^2}_{grid,b,y}} \tag{12}$$

and the local thrust in the grid points are calculated

$$dT_{l,b} = \frac{1}{2} \rho \, U^2_{r,b} \, C_{y,b}(\alpha_b) \, c \, N_B \tag{13}$$

where $C_y(\alpha)$ is the lift and drag coefficient projected into the axial direction.

The computation of the local torque is done in the same manner. Then the two resulting thrust and torque coefficients are interpolated based on the azimuth angle $\Psi$ of the two blades $b = 1,2$ and the azimuth angle of the grid point:

$$C_{T/Q,grid} = C_{T/Q,1} + (\Psi_{grid} - \Psi_1)\frac{(C_{T/Q,2} - C_{T/Q,1})}{\Psi_2 - \Psi_1} \tag{14}$$

## 3.5 Yaw modeling

It is evident that skewed inflow to the disc violates the conditions for the basic momentum equation (1) so that the momentum considerations used for derivation of the model are no longer valid. When the rotor operates in yaw there are two main effects on the induced velocities as described by Glauert (1935). One effect is the change in the mean level of the induced velocities and the other effect is an azimuth variation of the induced velocities as the wake vortex system is relatively closer to the rotor

on the one side compared to the other side.

A comprehensive investigation of yaw and dynamic inflow models for wind turbines and dynamic inflow modeling was carried out in the EU funded project "Joint Investigation of Dynamic Inflow Effects and Implementation of an Engineering Method" (Schepers and Snel, 1995). Here also a short summary of yaw models for helicopters is presented as these classical yaw models have been the basis for yaw models for wind turbines. The derivation and tuning of the present yaw model deviates slightly in

the way that AD simulations of a uniformly loaded disc are used where cylindrical vortex models were a main source in the project (Schepers and Snel, 1995). However, as the AD and vortex models should give almost the same results we will see that the present yaw model is close to some of the models from the above mentioned Dynamic Inflow EU project.





### 3.5.1 Mean induction in yawed inflow

The general equation relating the thrust and induction at a rotor operating in yaw, see Fig. 6, as proposed by Glauert (1935):

$\quad T = \rho A |\boldsymbol{U}_0 + \boldsymbol{u}_i| (-2\, u_{i,y})$ (15)

The equation has not been proven but is generally accepted as a good assumption and commonly used in helicopter AD modeling (Stepniewski and Keys, 1984). Now the following equation relating the thrust coefficient to the induction can be derived:

$$C_T = 4a(1 + a^2 - 2a\cos\Phi)^{\frac{1}{2}}$$ (16)

$\quad$ where $\Phi$ is the yaw angle.

Based on these results a reduction factor $k_a$ for the induction $a$ as function of $C_T$, Eq. (2) for different yaw angles can be derived:

$$a = k_a(k_3 C_T^3 + k_2 C_T^2 + k_1 C_T)$$ (17)

$\quad$ This reduction factor is approximated by a polynomial fit of the form

$$k_a(C_{t,mean}) = k_{a,3}\min(C_{t,mean}, 0.9)^3 + k_{a,2}\min(C_{t,mean}, 0.9)^2 + k_{a,1}\min(C_{t,mean}, 0.9))$$ (18)

as shown in the left plot of Fig. 5. The values of $C_{t,mean}$ used in Eq. (18) have to be limited to a maximum value of 0.9 to avoid a bending over of the $C_T(a)$ curve. The resulting approximation of the $C_T(a)$ curve is compared to Eq. (16) in the right plot of Fig. 5. The agreement becomes very good for low loading ($C_T < 0.9$) but becomes worse for higher loading. At higher loading,

$\quad$ however, Eq. (16) might not any longer be valid which justifies the limiter in Eq. (18).

The parameters $k_{a,i}$ of Eq. (18) are approximated as function of the yaw angle:

$$k_{a,i} = k_{i3}\Phi^3 + k_{i2}\Phi^2 + k_{i1}\Phi$$ (19)

The values $k_{i,j}$ are collected in the Matrix $\mathbf{K}$:

$\quad \mathbf{K} = \begin{bmatrix} 2.0705 & 2.1667 & -0.6481 \\ 2.1735 & 2.6145 & 0.8646 \\ 0.5136 & 0.4438 & -0.1640 \end{bmatrix}$ (20)





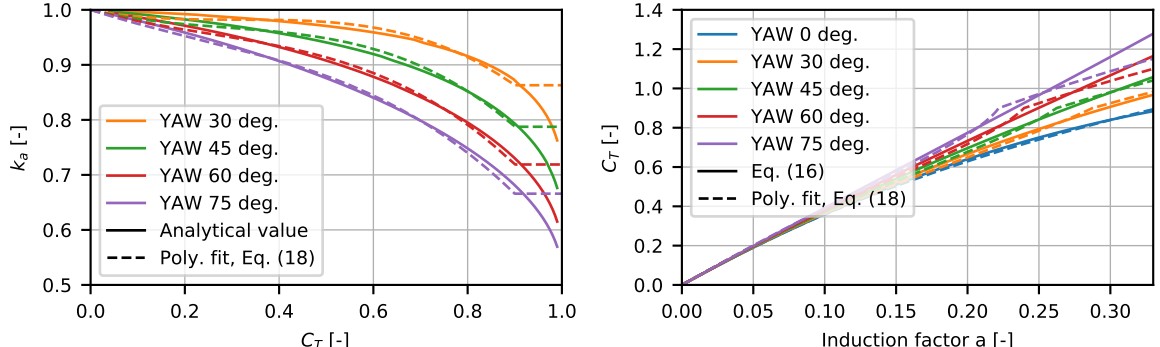

**Figure 5.** Left: Figure showing the reduction factor of the induction $a$ as function of $C_T$ for different yaw angles. Right: Relation between the thrust coefficient $C_T$ and the induced wind speed factor $a$ in yawed inflow.

The wake skew angle $\chi$ is as default found based on the average wake angle using vectors $\bar{\boldsymbol{U}}_0$ and $\bar{\boldsymbol{u}}_i$ representing the average local wind speed and induction over the whole rotor, see also Fig. 6.

$$\tan(\chi) = \frac{|\bar{\boldsymbol{U}}_0|\sin(\Phi)}{|\bar{\boldsymbol{U}}_0|\cos(\Phi) - |\bar{\boldsymbol{u}}_{i,y}|} \tag{21}$$

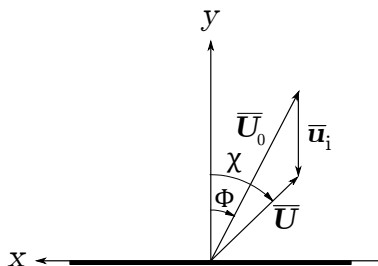

**Figure 6.** Top-view of the velocity vectors and angles used for the skew wake expression. The y-direction is the default wind direction without any skew inflow.

The wake skew angle $\chi$ depends on the thrust coefficient, which is illustrated in Fig. 7. At low loading $\chi$ is close to the yaw angle but for high loading it is seen that the wake can be deflected more than $10°$ from the mean wind direction.



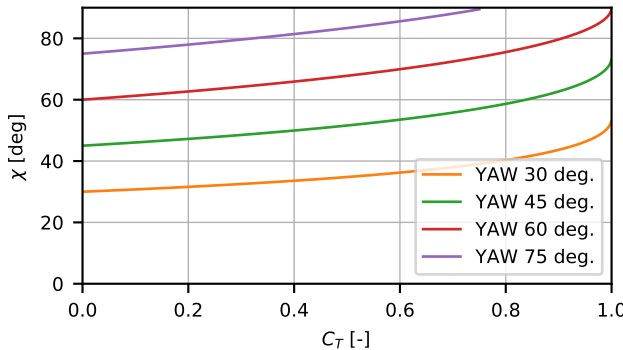

**Figure 7.** The wake skew deflection angle $\chi$ against the thrust coefficient for different yaw angles. For zero loading the angle is equal to the yaw angle, whereas the deflection angle increases in combination with an increased thrust level on the turbine.

### 3.5.2 Azimuthal variations of induction in yawed inflow

As the wake in the yawed conditions is skewed behind the rotor disc expressed by the skew angle $\chi$, see Fig. 6, the induction will be higher on the side of the rotor towards which the wake deflects. This is because the wake vortices are closer to the rotor
on that side.

A very general equation for the azimuthal variation of the induction was presented by (Schepers and Snel, 1995) containing a Fourier expansion in azimuth angle of the induced velocity at each radial position. We use here slightly simpler expression by Leishman (2005):

$$u_{i,y(\Psi)} = u_{i,y}(1 + k_x r \sin(\Psi) + k_z r \cos(\Psi)) \tag{22}$$

where $\Psi$ is the rotor azimuth, $r$ is the non-dimensional radius and $k_x$ and $k_y$ are constants.

Leishman (2005) has collected the values of $k_x$ and $k_y$ from several of the classical yaw models for helicopter rotors as shown in Table 1. It should be noted that these proposals are mainly thought for application on helicopter rotors in forward flight. As we will see below we found by comparison with results from an actuator disc in yaw that the best correlation was achived for

$$k_x = \tan(0.4\chi) \quad \text{and} \quad k_z = 0 \tag{23}$$

This is close to the model of Coleman as seen in Table 1.





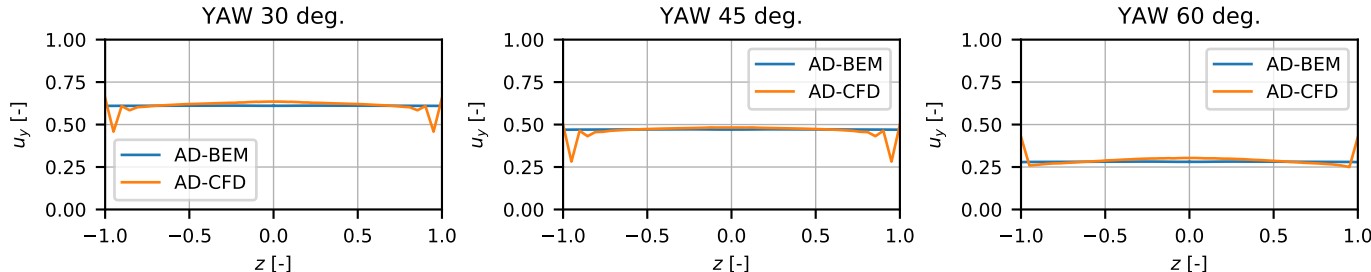

**Figure 8.** Comparison of axial velocity through a vertical line ($z$-axis in Fig. 6) through the rotor disc. The rotor loading is prescribed to a constant loading of $C_T = 0.8$.

**Table 1.** Coefficients for different yaw models (Leishman, 2005) extended and adapted to our coordinate system.

| Author(s) | $k_x$ | $k_z$ |
|---|---|---|
| Coleman et al. (1945) | $\tan(\chi/2)$ | 0 |
| Drees (1949) | $(4/3)(1 - \cos\chi - 1.8\mu^2)/\sin\chi$ | $-2\mu$ |
| Payne (1959) | $(4/3)(\mu/\lambda/(1.2 + \mu/\lambda))$ | 0 |
| White & Blake (1979) | $\sqrt{2}\sin\chi$ | 0 |
| Pitt & Peters (1981) | $(15\pi/23)\tan(\chi/2)$ | 0 |
| Howlett (1981) | $\sin^2\chi$ | 0 |
| Present | $\tan(0.4\chi)$ | 0 |

### 3.5.3 Comparison of the yaw model with Actuator Disc results

In the Fig. 8 and 9 the above described yaw model is compared with actuator disc results for a uniform, prescribed loading with a thrust coefficient of 0.8. In the BEM simulations the constant $C_T$ was prescribed as well.

As seen in Fig. 8 the axial wind speed distribution at the rotor disc is seen to match very well in the vertical plane (z-y plane), which clearly illustrates the good performance of Glauert's expression for the mean induction at different yaw angles. It should be noted the drop in velocity for the AD-CFD results closed to the edge is probably caused by the strong vorticity shed at the edge due to the jump in loading at the edge of the AD.

Results for the horizontal plane are depicted in Fig. 9 and the slope of the velocity variation across the disc is seen to correlate well between the AD and the BEM yaw model. However, towards the rotor edge the AD induction is higher on the side where

the wake is deflected to.

In summary, it can be concluded that the present yaw model is in close alignment with some of the models derived and presented in (Schepers and Snel, 1995). The Glauert correction for the mean induction seems to work very well which was





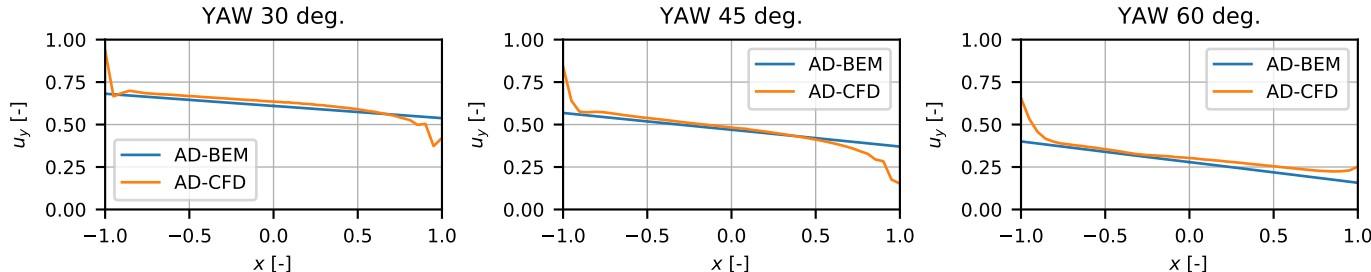

**Figure 9.** Comparison of axial velocity through a horizontal line through the rotor disc. The rotor loading is prescribed to a constant loading of $C_T = 0.8$.

also the conclusion in (Schepers and Snel, 1995). The azimuthal variation seems to be well represented by the Coleman model but we found the coefficient 0.4 on $\chi$ instead of 0.5, see Table 1. However, one major difference by the present model is that

implemented together with the grid BEM induction model we get a feedback on the induction from the yaw model which thus gives an additional azimuthal variation of the induction. This issue is addressed by (Burton et al., 2011) mentioning that a lack of feedback is a contradiction in the derivation of e.g. the Coleman model: constant loading (circulation) is assumed as a starting point but the solution is an azimuthal variation of induction and loading.

### 3.6 Dynamic inflow modeling

A time varying loading of the AD will cause a time delay of the velocities at the disc as the whole wake flow has to adapt to the new loading. This phenomenon, the dynamic inflow effect, was also part of the above mentioned EU funded project "Joint Investigation of Dynamic Inflow Effects and Implementation of an Engineering Method" (Schepers and Snel, 1995) where details about different modeling approaches can be found.

As for the yaw modeling we use again the AD-CFD model results to develop and tune an engineering sub-model for the

dynamic inflow. The AD simulations are carried out with a uniform loading and a step change in $C_T$ from 0.0 to 0.89 and another case with opposite loading sequence from 0.89 to 0.0. The computed axial velocities $u$ at the disc for different radial positions are shown in Fig. 10 as function of non-dimensional time t (time divided by $R/U_0$). It should be mentioned that the step size response was normalized to the BEM result of (1 to 0.666) for a change in $C_T$ of 0.89 for the different radial positions. This is because the AD-CFD results even for a constant loading typically show a non-uniform flow profile with the lowest

velocity of 0.655 at the tip and a higher velocity of 0.695 at the center. This non-uniform velocity profile for a constant loading has been found and discussed by several authors: Madsen, H. Aa. (1997), J. N. Soerensen and Munduate (1998), Madsen et al. (2010a) and van Kuik (2018).





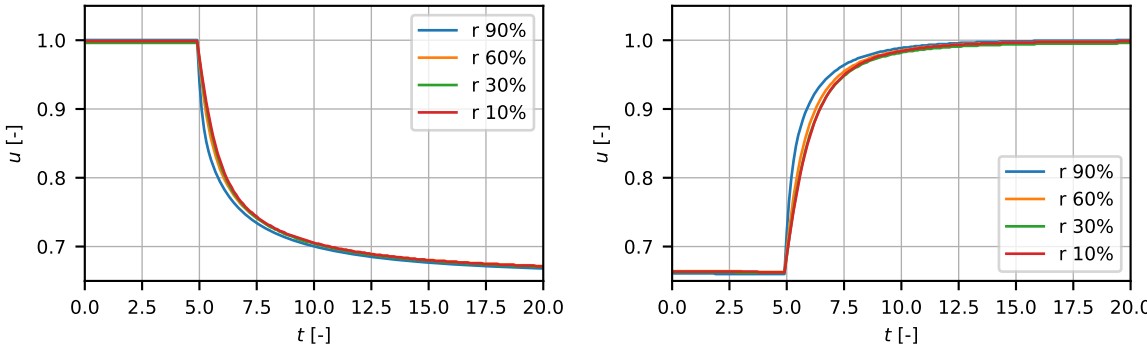

**Figure 10.** The response of the axial velocity to a step change in loading at the actuator disc at different radial position. In the left is the response with $C_T$ from 0.0 to 0.89, in the right is the response with $C_T$ from 0.89 to 0.0. The step change of loading is at $t=5$.

Comparing the decay in velocity for the different radial positions in the left graph of Fig. 10 it can be seen that the decay is slightly faster towards the tip than at the center. Likewise the increase in velocity for decreasing step loading is also slightly

faster at the tip as seen in the right graph of Fig. 10. The physical mechanism for this small difference in flow response along the radius is that the change in the constant loading sheds a vortex at the edge of the AD with strongest and fastest induction response in the edge region.

Approximating the response with an engineering model led to the conclusion that two time constants are necessary to obtain an accurate fitting to the AD data. We use the following expression for the two first-order filters:

$$u_{av}(t,r) = u(t,r) - \Delta u(t,r) \left[ A_1 \exp\left( -t \frac{f_1(a)}{\tau_1(r)} \right) + A_2 \exp\left( -t \frac{f_2(a)}{\tau_2(r)} \right) \right] \tag{24}$$

Here $u(t,r)$ is the flow speed at time $t$ at radius $r$, $u_{av}$ is the corresponding filtered flow speed, $A_1$ and $A_2$ are weighting constants of the two filters, $\tau_1$ and $\tau_2$ are the two time constants and finally $f_1(a)$ and $f_2(a)$ are functions that adapt the time constants to the local flow speed depending on the induction factor $a$. The functions take the form:

$$f_1(a) = (1 - k_{f1} a) \quad \text{and} \quad f_2(a) = (1 - k_{f2} a) \tag{25}$$

where $k_{f1}$ and $k_{f2}$ are constants.

We use a numerical optimization routine to find the set of parameters that minimizes the difference between the AD-CFD step response curves in Fig. 10 and the results of the model in Eq. 24. The variation of the two time constants along the radius are approximated with second-order polynomials in non-dimensional radius.

The optimization gave the following polynomials for the time constants:

$$\tau_1(r) = -0.7048\, r^2 + 0.1819\, r + 0.7329 \quad \text{and} \quad \tau_2(r) = -0.1667\, r^2 + 0.0881\, r + 2.0214 \tag{26}$$



The $\tau$ functions are shown in the left graph of Fig. 11. It can be seen that while the highest time constant shows almost no variation along the radial distance the lowest time constant decreases towards the tip which corresponds to the faster flow response towards the tip as described above.

A further result of the optimization is the weighting constants of the two filters which gave the following result:


$$A_1 = 0.5847 \quad \text{and} \quad A_2 = 0.4153 \tag{27}$$

Finally, the functions for the local flow speed to adjust the time constants were determined as:

$$f_1(a) = (1 - 0.50802\,a) \quad \text{and} \quad f_2(a) = (1 - 1.9266\,a) \tag{28}$$

This result shows that the highest time constant $\tau_2$ has to be scaled with a velocity very close to the wake flow velocity of $(1 - 2a)$ whereas $\tau_1$ is scaled with a flow velocity that is between the flow velocity at the rotor disc $(1 - a)$ and the free stream

velocity.

As a test case of the implementation of the above described dynamic inflow model implemented in the HAWC2 model we run the same prescribed variation of $C_T$ as used above to derive the time constants. The comparison of the AD and HAWC2 model results in Fig. 11 shows a very good correlation as should be expected. This is in good agreement with the work by Yu (2018) who derived a two time constant dynamic inflow model based on vortex models of an actuator disc.

In a time marching formulation the dynamic inflow filter at each grid point can be implemented as follows:

$$u_{i,y}^t = A_1 u_{i,y,1}^t + A_2 u_{i,y,2}^t \tag{29}$$

$$u_{i,y,1}^t = u_{i,y,1}^{t-1} e^{-\Delta t f_1(a)(\tau_1(r))^{-1}} + u_{i,y,QS}^t (1 - e^{-\Delta t f_1(a)(\tau_1(r))^{-1}}) \tag{30}$$

$$u_{i,y,2}^t = u_{i,y,2}^{t-1} e^{-\Delta t f_2(a)(\tau_2(r))^{-1}} + u_{i,y,QS}^t (1 - e^{-\Delta t f_2(a)(\tau_2(r))^{-1}}). \tag{31}$$

where the superscripts $t$ and $t - 1$ denote the present and previous time step and $u_{i,y,QS}$ is the quasi steady induced velocity

including the yaw correction, Eq. (22).

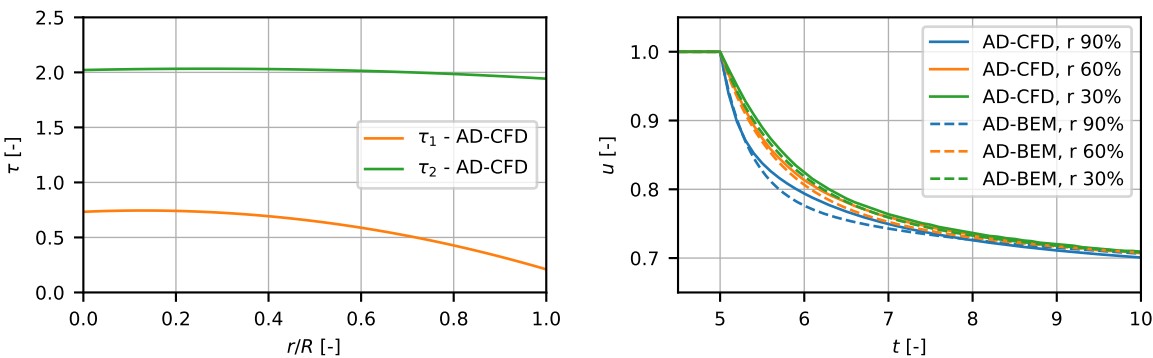

**Figure 11.** Left: Figure showing the derived time constants as a function of non-dimensional radius. Right: Comparison of the step response of the model using tuned constants with the AD-CFD simulations.



### 3.6.1 Summary on dynamic inflow

Comparing the present dynamic inflow model with the models derived and presented in (Schepers and Snel, 1995) we find again as for the yaw models close correlation. Firstly, the AD results clearly indicate that two time constants are needed where the highest constant is almost independent of radial position but the low one decreases towards the tip. The need of two

time constants was also found in (Schepers and Snel, 1995) using the cylindrical vortex models. Secondly, we find that the time constants need to be normalized with a local convection velocity which we found to be quite different for the two time constants. This was also the case for some of the models in (Schepers and Snel, 1995).

### 3.7 In-plane sweep and out-of-plane bending

For non-straight blades with sweep/prebend or in-plane and out-of-plane deflection the radial distance between adjacent grid

points is not equal to the distance along the curved blade. Therefore both $C_T$, Eq. (3), and $C_Q$, Eq. (4), have to be multiplied with $ds/dr$, the derivative of the blade span with respect to the radius.

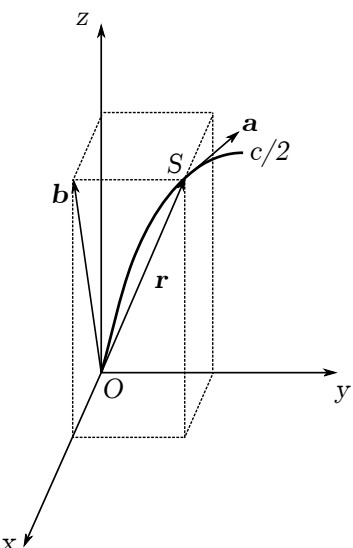

**Figure 12.** Sketch of a non-straight blade with in-plane and out-of-plane deflection.

The calculation of $ds/dr$ is demonstrated in the rotor coordinate system, see Fig. 12. The y-axis is pointing downwind and the theoretical BEM rotor disc is in the x-z plane. The curved blade is represented by the half-chord line. The vector $\boldsymbol{a}$ is tangent to the half chord line at this section point $S$. The vector $\boldsymbol{r}$ is pointing from the root point $O$ to the section point $S$. The

projection of vector $\boldsymbol{r}$ to the rotor plane (x-z plane) is $\boldsymbol{b}$. It is equivalent to setting the y-component of $\boldsymbol{r}$ to zero.





The curved length $s$ is increasing in the direction of $\boldsymbol{a}$, and the radius is increasing in the direction of $\boldsymbol{b}$. Thus, $ds/dr$ can be calculated as:

$$ds/dr = \frac{|\boldsymbol{a}||\boldsymbol{b}|}{|\boldsymbol{a} \cdot \boldsymbol{b}|} \tag{32}$$

### 3.8 Radial induction model

The standard BEM theory does not give information about the radial induction component and for plane rotors this induction component will only have minor influence on the loading. However, for rotors with out-of-plane bending blades or rotors with coning the radial induction component will have an impact on the angle of attack (AOA) and thus also on the loading. An analytical expression for the lateral induction for a 2D actuator disc is presented in Madsen, H. Aa. (1997) and is adopted for an axis-symmetric AD in Madsen et al. (2010a). The expression is:

$$v(r) = \frac{1}{2.24} \frac{C_{Tav}(r)}{4\pi} ln\left[\frac{0.04^2 + (r+1)^2}{0.04^2 + (r-1)^2}\right] \tag{33}$$

where $C_{Tav}(r)$ is the mean thrust coefficient as function of radial position defined as:

$$C_{Tav}(r) = \frac{\int_0^r C_T(r)\, 2\,\pi\, r\, dr}{\pi\, r^2} = 2\frac{\int_0^r C_T(r)\, dr}{r} \tag{34}$$

where the local thrust coefficient $C_T(r)$ is given in Eq. (3). The use of $C_{Tav}(r)$ instead of the total thrust coefficient is important only when $C_T$ shows a strong variation as function of radial position.

We test the radial induction model by a comparison with the AD-CFD solution for a constant loading of $C_T = 0.89$. As seen in Fig. 13 the radial induction computed with the engineering sub-model correlates very well with the AD-CFD result.

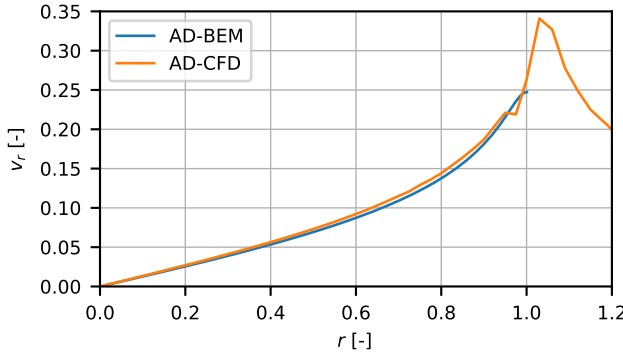

**Figure 13.** The radial induction computed with an engineering sub-model in comparison with AD-CFD result for a constant loading with a thrust coefficient of 0.89.



## 3.9 Overview of the model

An overview of the complete aerodynamic model is shown in Algorithm 1. The algorithm includes references to the relevant equations in this article and can be used as a manual for implementation of the grid BEM algorithm. It is crucial that the dynamic inflow filter is applied at the very end of the algorithm to prevent nonphysical rapid induction changes due to any of the sub-models. Otherwise, for example, a change in yaw angle at one time instant at the rotor disc in turbulent inflow would lead to an immediate change of the induced velocities, even though the wake did not have time to deflect.

For a typical set-up we use 16 azimuthal grid points. The number of radial grid points are somewhat dependent on the planform and tip shape but typically 30-50.

The aerodynamic model as described here is the aerodynamic model in HAWC2. However, it is also found in a stand-alone version HAWC2_Aero which can run the same type of simulations with turbulent inflow, pitch actions and rpm variations as HAWC2 but for a stiff structure. In this version the simulation speed with all input/output operations is in the order of 7-10 times real time. This means that the computational time for the aerodynamic part is still small (10-20 %) relative to the total computational time for the aeroelastic simulations although we, in this BEM implementation, update the induction over the whole disc at each time step. One reason for this is that no sub-iterations in the induction modeling are necessary.

Unsteady airfoil aerodynamics effects (dynamic stall and Theodorsen effects in attached flow) are not included in the computation of the induced velocities. This is possible because unsteady airfoil aerodynamics occur at much faster time scales with time constants that depend on the half chord divided by the relative speed. For comparison, the dynamic inflow time constants scale with the rotor diameter divided by the free wind speed. After the induced velocities are computed, the unsteady airfoil aerodynamics are determined using the Beddoes-Leishman-type model described by Hansen et al. (2004), which was recently extended by Pirrung and Gaunaa (2018).

## 4 Turbulent inflow computations

In this section we demonstrate the impact of the present grid BEM implementation on the induction and load characteristics based on simulations of the AVATAR rotor (Sieros et al., 2015) in turbulent inflow. The impact is evaluated by comparing with an "annular mean BEM" version computing the mean induced velocities in an annular element. This annular mean BEM version was incorporated in a test version of HAWC2 for the present investigation.

### 4.1 The induction mechanism for turbulent inflow

The induction mechanism simulated with the grid BEM implementation for turbulent inflow is illustrated in Fig. 14. The simulations were carried out on the AVATAR turbine (Sieros et al., 2015) with a 205 m diameter rotor at $10 \text{ m s}^{-1}$ and a turbulence intensity of 15 %, no shear and constant rotor speed. In the left graph of Fig. 14 we show the induced velocities at 4 grid points on the stationary rotor grid at radius 42.5 m with the monitoring points shown on the grid to the right. The induced velocities can be seen to vary slowly in time. They can be quite different in some periods due to the large turbulence scales



---

**Algorithm 1** Overview of the aerodynamic model

---

**for** each time step **do**

    **Yaw modeling:**

        Compute rotor mean induction

        Compute rotor mean wind speed

        Compute wake skew angle $\chi$, Eq. (21)

        Compute $k_x$ and $k_y$, Eq. (22)

    **for** each radial grid point **do**

        **for** each azimuthal grid point **do**

            Find the two closest blades

            **for** each of the two closest blades $b = 1, 2$ **do**

                Compute $ds/dr$, Eq. (32)

                Calculate thrust coefficient $C_{T,b}$, Eq. (3), and torque coefficient $C_{Q,b}$, Eq. (4), using:

                    1) from blade section: section velocity, deflection, pitch angle, twist

                    2) at grid point: induced wind and free wind

                Divide $C_{T,b}$ by the tip loss factor $F$, Eq. (7)

            **end for**

            Interpolate $C_T$ and $C_Q$ based on azimuth angle of the two closest blades

            Compute induction factor $a$ based on polynomial, Eq. (2)

        **end for**

    **end for**

    Compute rotor average thrust coefficient (for mean induction yaw correction)

    Compute $k_a$ for mean induction yaw correction, Eq. (17)

    **for** each radial grid point **do**

        **for** each azimuthal grid point **do**

            $a = a k_a$

            Compute $a'$, Eq. (6)

            Compute induced velocities using wind speed and $\Omega r$ at grid point

            Compute radial induction, Eq. (33)

            Apply azimuth varying yaw correction, Eq. (22,23)

            Apply dynamic inflow filter, Eq. (29)

        **end for**

    **end for**

    **for** each blade section **do**

        Compute aerodynamic forces including dynamic stall and theodorsen effects (Hansen et al., 2004; Pirrung and Gaunaa, 2018)

    **end for**

**end for**

---




causing different inflow velocities over the rotor. However, the induction seen from the rotating blade varies considerably faster as it samples the induced velocities at the different azimuth grid positions. This rotational sampling of the induction field is

thus basically the same mechanism as the rotational sampling of turbulence.

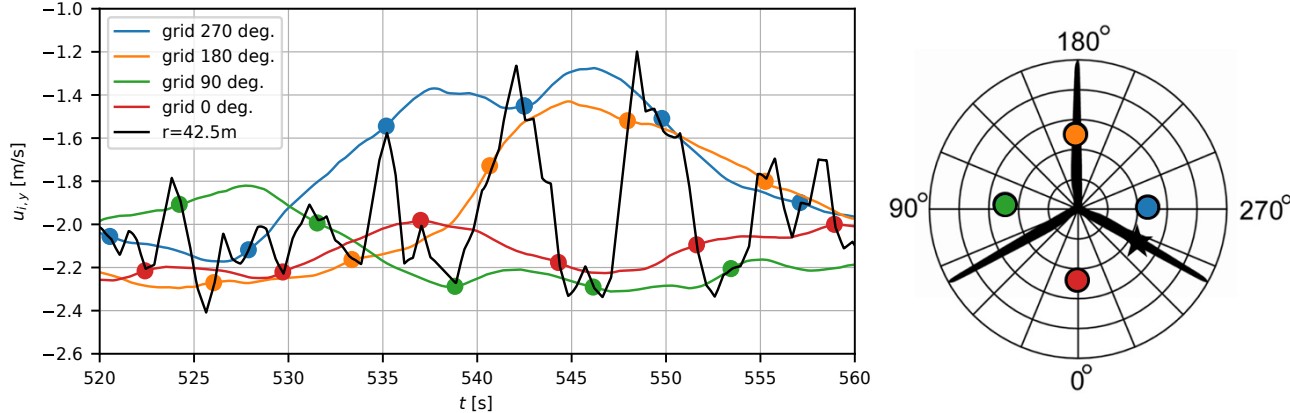

**Figure 14.** Illustration of the dynamic induction mechanism in turbulent inflow showing the blade scanning through the field of slow varying induction velocities but transferring to higher frequencies due to the rotational sampling of the turbulence.

## 4.2 Characteristics of the induced velocities

To illustrate further the characteristics of the induced velocities from the AVATAR rotor case mentioned above the time trace of the induced velocity at radius 43 m is shown in the left graph of Fig. 15. Further is shown for comparison the induced velocity simulated with the annular mean BEM method. The dynamic characteristics are clearly completely different which is further

explored by the PSD spectra shown in the right graph of Fig. 15. The spectrum of the induced velocity computed with the grid BEM model have distinct peaks at 1P, 2P etc. and can be seen to have close resemblance with the spectrum of the axial free wind speed component relative to the blade at same radial position. As the rotational sampling of both the inflow and the induced velocity field has the same characteristics, it indicates that the induced velocity field over the rotor also has the same overall characteristics as the turbulent inflow, although considerably lower wind speeds.

As expected the PSD of the induced velocity computed with the annular mean method has no peaks and has some resemblance with the PSD of the hub wind speed.



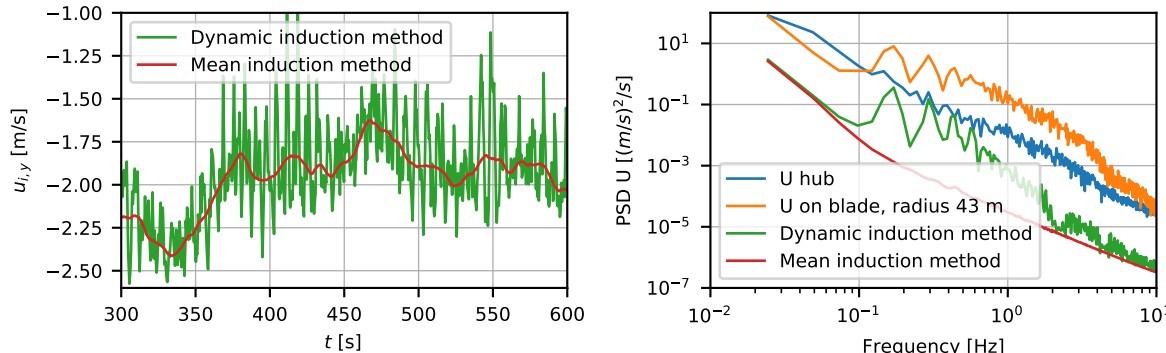

**Figure 15.** Left figure: Time traces of induced velocity at radius 43 m simulated with the annular mean and grid BEM method. The right figure shows the PSD of the same two traces. Additionally in the same figure the PSD of the free wind speed relative to the blade at the same radial position and of the hub wind speed is shown.

## 4.3 Load impact mechanism of the grid BEM induction method

We will now illustrate the mechanism behind the load impact of using a mean annular BEM approach and a grid BEM model, respectively. Again it is a simulation example for the AVATAR rotor.

A simulation was run with a ramp in wind speed from 4 to 20 m s$^{-1}$ for uniform inflow. The induced velocities at three radial positions on the blade are shown in Fig. 16. As the inflow is uniform both BEM implementations give the same result.

Now a simulation is performed for sheared inflow with an exponent of 0.5 and at a wind speed of 8 m s$^{-1}$ and 14.5 m s$^{-1}$, respectively. We show the induced velocity for the same radial positions on the blade as function of the local inflow velocity at that point.




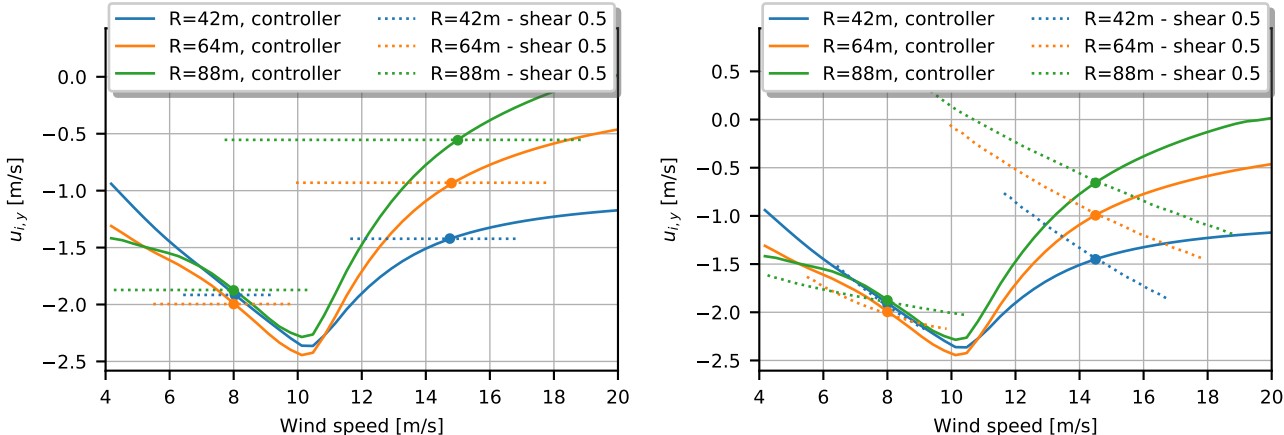

**Figure 16.** The solid lines in both figures are the computed induction at three radial positions for the AVATAR rotor for a wind speed ramp from 4 m s$^{-1}$ to 20 m s$^{-1}$ for uniform inflow and normal controller with variable speed and pitch regulation. The dashed lines are now the azimuthally varying induced velocities at the same radial positions for operation in sheared inflow with an exponent 0.5 and at a mean wind speed of 8.0 and 14.5 m s$^{-1}$,respectively. In the figure to the left the results for the mean annular BEM model is shown and in the right figure it is the grid BEM model results.

For the mean annular BEM the constant induced velocity as function of the local wind speed on the blade is obvious. The mean value might be slightly different from the value at the same wind speed for the turbine operating in uniform inflow due to non-linear effects from computation of the mean loading.

The picture is quite different for the grid BEM method as shown in the figure to the right. For all radial positions at both wind speeds we see that the induced velocity increases in magnitude but with the steepest slope at high wind. The mechanism behind this is that as soon as the inflow velocity is different from the hub wind speed the local blade section operates in conditions

where either the rotational speed or/and the pitch do not correspond to the equilibrium conditions for that wind speed.

At 8 m s$^{-1}$ it's mainly the rpm that influences the variation in the local induction around the mean operational wind speed of 8 m s$^{-1}$. When the local wind speed is lower than 8 m s$^{-1}$ the local tip speed ratio is above the mean value and the blade section operates at a higher thrust coefficient. The opposite holds when the local wind speed is above the mean wind speed. The result

is that the relation of the induction versus wind speed deviates from the induction curve for the turbine in uniform inflow. It also appears that the slope of this local relationship between induced wind speed $u_{i,y}$ and free local wind speed $U_0$ decreases from root to tip. For local wind speeds below the operational point, the increased $C_T$ will increase the induced wind speed whereas the decreased local wind speed (being a factor on the induction) will decrease the induced wind speed. In most conditions, the impact of the local wind speed multiplied on the induction factor is strongest but at high thrust coefficient regions towards the

tip and for bigger deviations from the mean wind speed we can see that the $C_T$ impact increases and the slope of the $u_{i,y}(U_0)$ curve decreases. The slope can even be positive for very high thrust coefficients.

At 14.5 m s$^{-1}$ it's mainly the pitch that influences the induction variation around the mean wind speed as the rpm is constant





for wind speeds above rated. So when the blade section is in a region with a lower wind speed than the mean, the pitch is too high which gives a lower thrust and a reduced induction. Opposite again when the local wind speed at the blade section is

above the mean, the pitch is too low corresponding to that wind speed which gives an increased induction. In this region we can thus conclude that both the effect from the changes in $C_T$ and the variation in local wind speed when deviating from the mean operational point have the same sign which means that we always will see a decreasing induction from a decreasing local wind speed and vice versa for a local wind speed above the mean operational point.

The important impact on the loads is that changes in the local wind speed will always be counteracted to some extent by the

induced wind speed and thus reduce the variations in AoA and likewise variations of the aerodynamic loads. This will be further explored below for turbulent inflow and quantified for a few test cases.

### 4.4    Induced velocities for turbulent inflow

The characteristics of the induced velocities for turbulent inflow are basically determined by the same mechanism as described above for sheared inflow. As discussed above the turbulent inflow with dimension of structures less than one rotor diameter

cause a non-uniform inflow over the rotor disc. It means as for sheared inflow that a point on the rotating blade will see a local wind speed different from the mean wind speed corresponding to the mean operational conditions of the turbine. In Fig. 17 the induced velocity at radius 64 m is shown as function of wind speed for uniform inflow. The induced velocity as function of local wind speed from the same position on the blade for simulation with turbulent inflow at a mean wind speed at 14.5 m s$^{-1}$ and a turbulence intensity of 15 % with standard control is shown as dots. In the left figure, it is for a mean annular BEM and

the grid BEM results in the right figure. As the mean wind speed changes continuously for turbulent inflow the $u_{i,y}(U_0)$ curves as discussed above are more difficult to see here. However for the mean annular BEM the horizontal patterns of the dots are visible. For the grid BEM we have to imagine that the $u_{i,y}(U_0)$ curves have the negative slope as shown above for shear at 14.5 m s$^{-1}$.





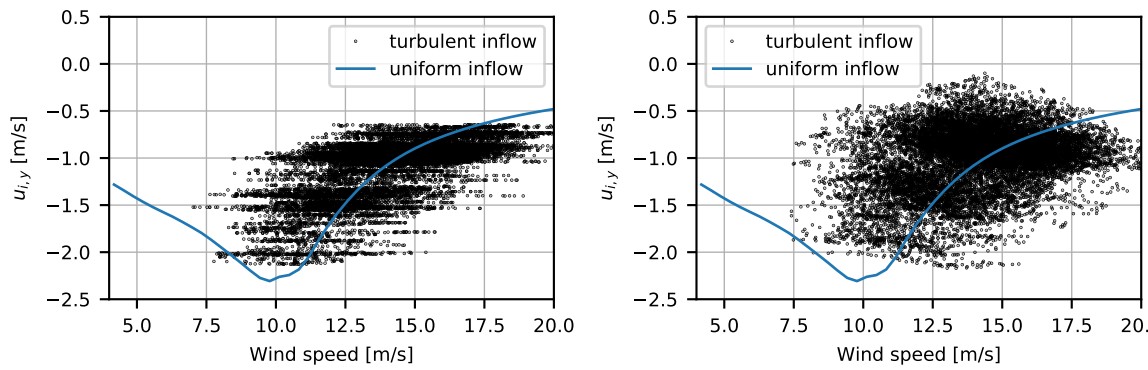

**Figure 17.** The solid lines in both figures are the computed induction at the radial position of 64 m for the AVATAR rotor for a wind speed ramp from 4 m s$^{-1}$ to 20 m s$^{-1}$ for uniform inflow and normal controller with variable speed and pitch regulation The dots are the induced velocities for turbulent inflow without shear for a mean wind speed of 14.5 m s$^{-1}$. In the figure on the left results from the mean annular BEM are shown, on the right results from the grid BEM.

### 4.5 Load and power impact for DLC 1.2 for the AVATAR and DTU reference wind turbine

The impact of the grid BEM model on fatigue loads and power production according to DLC 1.2 (IEC, 2005; Hansen et al., 2015) has been investigated. Computations were performed for both the DTU 10 MW reference wind turbine (RWT) (Bak and Zahle, 2013) and the AVATAR 10 MW turbine (Sieros et al., 2015). To avoid seed dependency, 18 seeds at each wind speed were used: 6 seeds at 0° yaw error and 6 seeds at ± 10° yaw error, respectively. The wind speeds range from 4 to 26 m s$^{-1}$ with a 2 m s$^{-1}$ spacing.

For brevity, this section focuses only on the 1 Hz equivalent load of the flapwise blade root bending moment and the mean power. All results are presented as percent relative difference compared to an annular BEM model that includes the yaw correction presented in Sect. 3.5. Results from a mean annular BEM model without yaw correction are also included so that the influence of grid versus mean annular BEM can be compared to the impact of a more widely used type of BEM model. To isolate the reaction of the induction model to shear and turbulence, additional runs of DLC 1.2 without shear and turbulence

are shown. The runs without shear use the same 18 seeds per wind speed at 0° , +10° , -10° yaw error as the regular DLC 1.2 computations.

The results for the DTU 10 MW RWT are shown in Fig. 18. It can be seen that the difference of grid compared to annular BEM has a much larger impact on the results than the yaw correction. The yaw correction has some influence at wind speeds below rated, but above rated the influence is close to zero.

Overall, the grid BEM results in significant lower fatigue loads, up to 8 %, except in a narrow wind speed interval between 7 and 10 m s$^{-1}$ with an increase of 1 % as seen in the upper, left graph of Fig. 18. When splitting up in contributions from turbulent inflow and shear we can see in the upper, middle graph that the fatigue from turbulence is reduced for all wind speeds for the grid BEM with a reduction of roughly 6 % at 16 m s$^{-1}$ and above. However, for the impact from shear the fatigue is

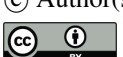

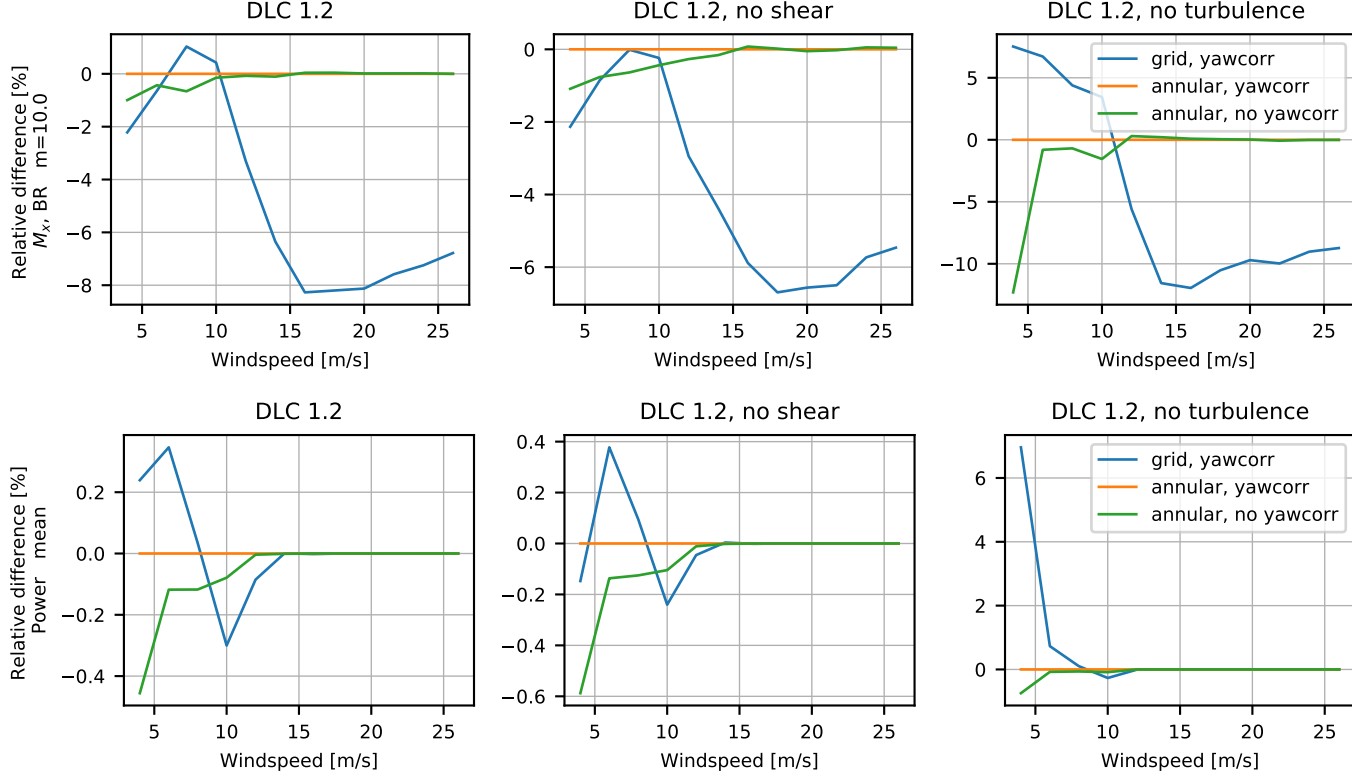

**Figure 18.** A comparison for the DTU 10 MW RWT of difference in blade root flapwise fatigue loads (top row) and mean power production (bottom row) for DLC 1.2 between the annual mean BEM method with and without yaw correction and the grid BEM version with yaw correction.

increased up to 6 % at low wind speeds which is due to the high thrust coefficient for that rotor causing a positive slope of the
$u_{i,y}(U_0)$ curve as discussed above.

The influence of the grid based BEM for the power production of the DTU 10 MW is very small at roughly $\pm 0.3$ % below rated. In the pure shear case at 4 m s$^{-1}$ a large power increase by 6.5 % can be seen, but that increase almost disappears for combined turbulent and sheared inflow.

The results for the AVATAR turbine, Fig. 19, show a much larger impact of the grid BEM approach, while the yaw correction
only has very minor influence. Relative to the annular BEM, the fatigue loads predicted by the grid BEM in pure shear are reduced on average by roughly 12 %, the loads in pure turbulence by 7.5 % and in the combined case by roughly 10 %. At the same time, the power below rated is predicted to increase by roughly 0.5 %, which seems to be mainly due to better operation in turbulent inflow.

Comparing the two cases we can conclude that the impact of the grid BEM approach depends on the actual turbine design with



an increasing reduction of fatigue loads for lower loaded (low induction) rotors. For both turbine designs the load reduction is considerable (8 to 10 %) for wind speeds above rated power.

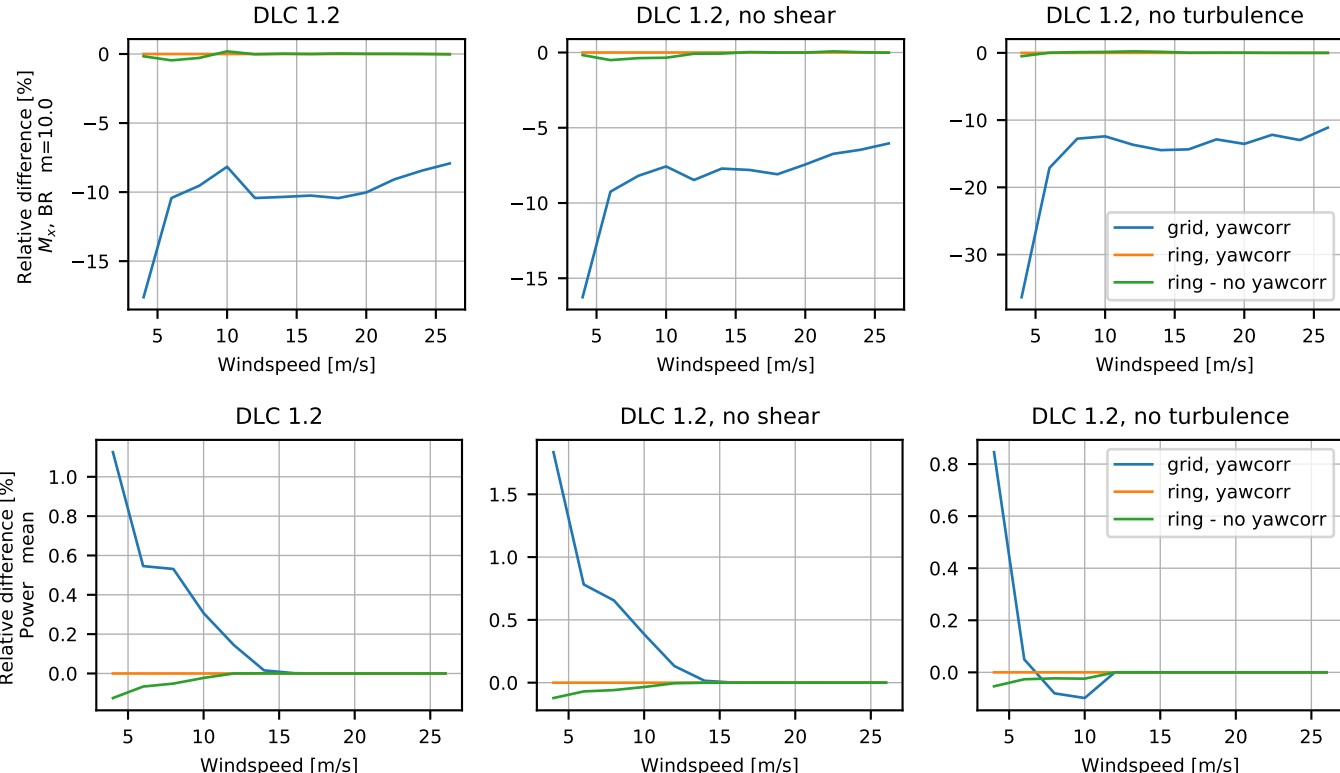

**Figure 19.** A comparison for the AVATAR turbine of difference in blade root flapwise fatigue loads (top row) and mean power production (bottom row) for DLC 1.2 between the annual mean BEM method with and without yaw correction and the grid BEM version with yaw correction.





## 5 Validation results

We present in this section a selection of validation results in order to illustrate the performance of the grid BEM implementation for different challenging inflow cases. As mentioned above the grid BEM method is the aerodynamic model in HAWC2 and the cases are simulated with this model. It also means that several validation cases can be found in different articles published in the past and only two of them are explicitly summarized here. The first referenced validation paper contains not only a validation of the aerodynamic model of HAWC2 but of the full aeroelastic model. However, in the second validation reference, the aerodynamic model in HAWC2 is alternated between the grid BEM and full 3D CFD which enables a detailed validation of the grid BEM results.

### 5.1 Published validation cases

In Larsen et al. (2011), a validation study of both the HAWC2 model and the DWM wake model (Larsen et al., 2008; Madsen et al., 2010c) was carried out on the basis of comparisons of model predictions with full-scale turbine measurements from the Dutch wind farm Egmond aan Zee consisting of 36 Vestas V90 turbines. In the paper, it is concluded that the measurements are of a remarkable high quality enabling comparison of not only fatigue loads, but also simple statistics in terms of maximum, minimum and mean values. It was found that when comparing the predicted power curves with measurements in both free and wake sectors, an excellent agreement is seen. Further, a very fine agreement was also seen between measured and simulated loads in both the free sector and a sector with wake effects from five turbines separated with seven diameters.

In the other validation publication by (Heinz et al., 2016) the coupling of the HAWC2 structural model to EllipSys3D is presented. This provides an excellent basis for validation of the grid BEM aerodynamic model for simulations on the NREL 5 MW turbine (Jonkman et al., 2009) as a direct comparison with high fidelity model results for the exact same input data and structural model is possible. Besides results for uniform inflow a comparison of flapwise and edgewise tip deflection as function of azimuth is presented for $0°$, $30°$ and $60°$ yaw angle. In the paper, it's concluded that: "Both models still show a very good agreement". Finally, a challenging case of an emergency shut down is presented and also for that case it is concluded that the responses of the two models agree very well.

### 5.2 Half wake

The first validation case is to demonstrate the model response to a considerable shear in the inflow for the NREL 5 MW turbine (Jonkman et al., 2009). We have chosen a case with shear in the horizontal plane because a vertical shear representing atmospheric inflow with shear can be considerably influenced by the interaction of the flow with the ground surface and thus disturbing the direct impact of the induction modeling (Madsen et al., 2010a). An artificial shear inflow was created changing the inflow velocity from 10 m s$^{-1}$ to 5 m s$^{-1}$ over a narrow region around the hub center, according to the analytical expression:

$$U_0 = U_{0,max} \left( 0.75 - 0.25 \left( tanh \left( 8.78044 \frac{x}{R} \right) \right) \right) \tag{35}$$





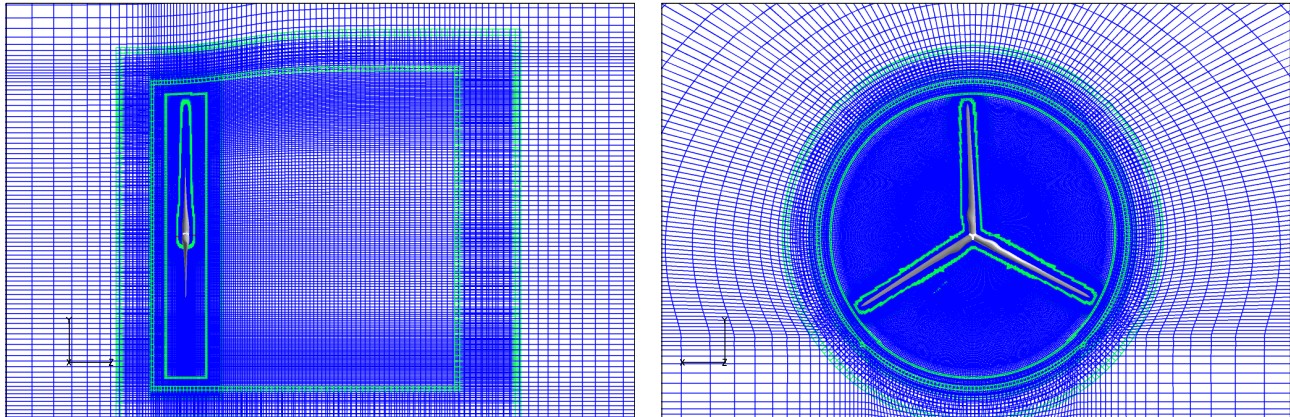

**Figure 20.** Side-view and front-view of the CFD mesh around the NREL 5 MW reference turbine generated with a hub height of 90 m.

where $x$ is the horizontal distance from the rotor center, and $R$ is the rotor radius. The resulting horizontal shear profile is shown in Fig. 21.

The CFD simulations were carried out with the 3D incompressible Navier-Stokes solver EllipSys3D by Michelsen (1992, 1994) and Sørensen (1995), with a surface resolved representation of the rotor, omitting the nacelle and tower. The flow on the no-slip surface of the rotor was assumed fully turbulent, and modeled using the $k - \omega$-SST model by Menter (1993). In the computations we used an overset grid approach in which a curvilinear rotor resolved mesh rotated together with two successively coarsened cylindrical meshes resolving the near field around the rotor, which were embedded in a larger stationary

coarse semi-cylindrical mesh resolving the far-wake with the far-field boundaries placed 8 rotor diameters away from the surface and a ground boundary modeled using a symmetry boundary condition placed 90 m below the rotor center. The surface of each blade was resolved with 256 cells in the chord-wise direction, 128 cells in the span-wise direction and grown into a volume mesh with 64 cells normal to the surface using the in-house hyperbolic mesh generator HypGrid Sørensen (1995). The first cell height was set to $1 \times 10^{-6}$ m resulting in a $y+$ value below 2. The full grid assembly contained $15 \times 10^6$ cells. Figure

20 shows a front-view and side-view of the volume mesh.

To minimise the computational time, both grid sequencing and time step sequencing were used. To settle the overall induction field the flow was simulated with a coarse time step of $2.2765 \times 10^{-2}$ corresponding to 300 time steps per revolution for 20 revolutions on a mesh coarsened by a factor of two in each coordinate direction (Gr3). With the same mesh refinement level, the time step was subsequently refined by a factor of five to $4.553 \times 10^{-3}$ yielding 1500 time steps per revolution for another

15 revolutions (Gr2). Finally, the mesh was refined to the finest grid level and the time step refined by a factor of two to $\Delta t = 2.2765 \times 10^{-3}$ (Gr1). The resulting mean integral forces of the grid/time step sequence is shown in Table 2.

A user defined shear flow can be input to a HAWC2 simulation so the case could be simulated by a default set-up. When comparing the normal and tangential loading on the blade at azimuth positions of $90°$ and $270°$ ($0°$ is vertical upwards) which are in the extreme low and high inflow regions we find overall a good correlation as can be seen in Fig. 22. There are minor

deviations in the tip region with where the grid BEM overestimates the normal force loading. Also the tangential loading is





| Grid | Torque (kNm) | Thrust (kN) |
|------|--------------|-------------|
| Gr3  | 2079.95 (5.8 %) | 353.23 (2.1 %) |
| Gr2  | 2032.72 (3.4 %) | 344.95 (0.3 %) |
| Gr1  | 1966.78 (-) | 346.07 (-) |

**Table 2.** Grid/time step convergence of the ElliPSys3D simulation, showing mean integral forces computed for the velocity step case at each of the three grid/time step levels.

slightly underestimated on the central part of the blade for the 270° azimuth position.

The case is further analyzed by comparing the integrated normal and tangential blade forces as function of azimuth as shown in Fig. 23. Again an overall good correlation between the high fidelity CFD results and the grid BEM results is found. However, there is a time delay for HAWC2 in the rising of the loads from low to high wind inflow (high to low $C_T$) around an azimuth
of 180 deg. However, the same is not seen at around 0° where the wind speed is changing from a high to low value.

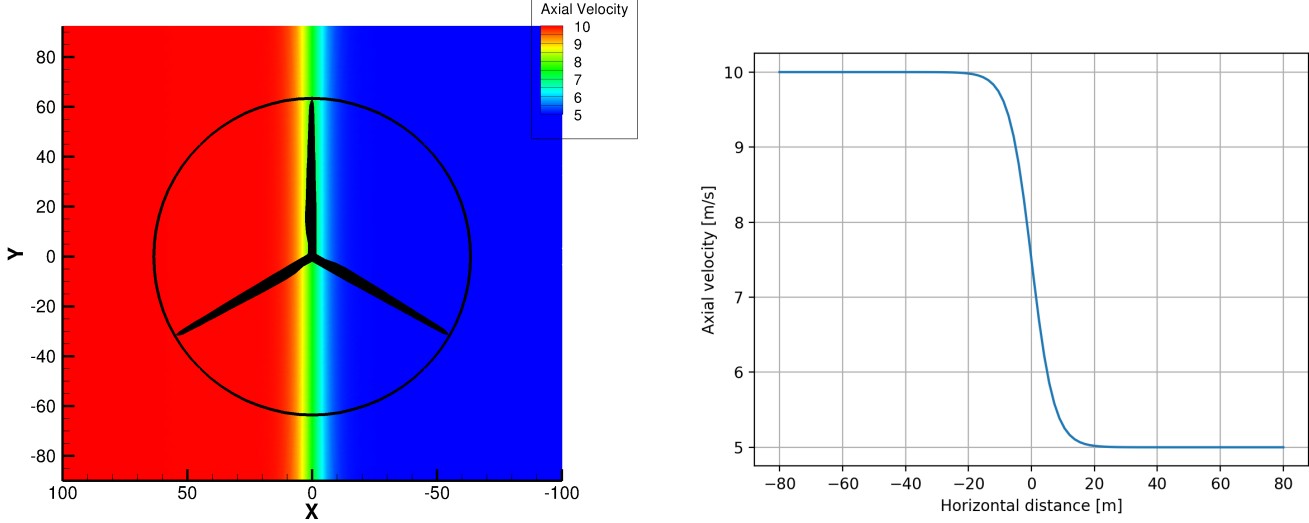

**Figure 21.** The graph shows the contour plot for the velocity field for the sheared inflow case.




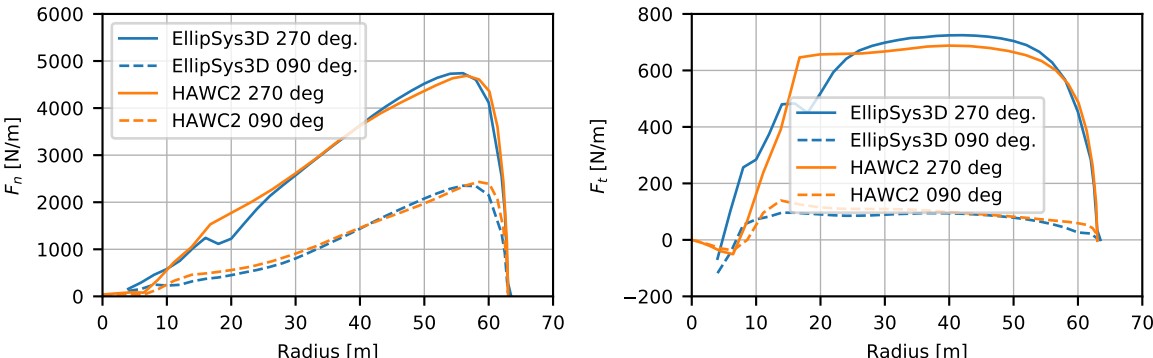

**Figure 22.** The normal force (left) and tangential force (right) computed with HAWC2 at two azimuth positions in comparison with Ellip-Sys3D results for half wake inflow.

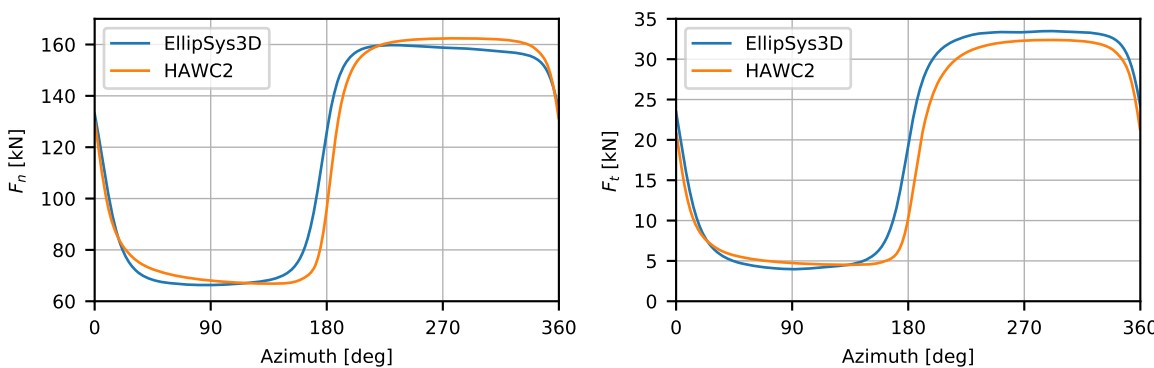

**Figure 23.** Left figure: A comparison of the normal force $F_n$ as function of azimuth computed with CFD and BEM, respectively. Right figure: Same comparison for the tangential force $F_t$.

### 5.3 Turbulent inflow

Detailed aerodynamic measurements on full-scale turbines are very limited. However, in the DanAero project such measurements were carried out in 2009 on a NM80 turbine with an 80 m diameter (Madsen et al., 2010b; Troldborg et al., 2014). The experimental set-up comprised surface pressure measurements at four radial positions from which the aerodynamic forces normal and tangential to the local cord were derived. A validation exercise using these data were described and presented recently by (Madsen et al., 2018) so we will only present a single set of results from that paper. The case is for a mean wind speed of 6.1 m s$^{-1}$, a turbulence intensity of 6.8 % and minimal shear. For details of the experimental and modeling set-up, the reader should see (Madsen et al., 2018).

The comparison of PSD spectra of the measured and simulated aerodynamic force perpendicular to the chord is shown in Fig.


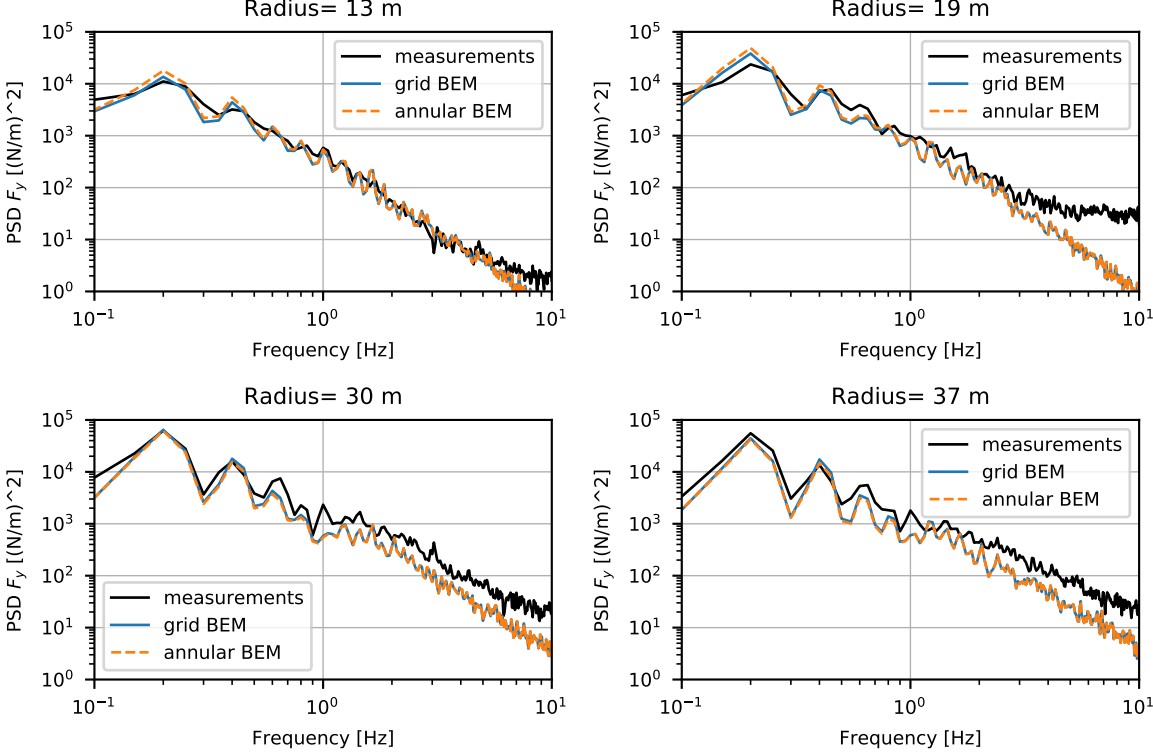

**Figure 24.** A comparison of HAWC2 simulations on the NM80 turbine and experimental results from the DanAero project. Power spectra of the chordwise aerodynamic force component parallel to the chord in comparison with measured results. Wind speed 6.1 m s$^{-1}$, negligible shear and rpm = 12.3. Figure reproduced from (Madsen et al., 2018) and adding the annular mean BEM results.

24. Besides the grid BEM results, we have also added the mean annular results for comparison with the measurements. Overall the correlation between simulations and measurements is good. In particular the 1P, 2P and 3P peaks are captured well and both simulation and experiment show the increasing size of the peaks towards the tip of the blade due to the rotation sampling effect of the turbulent inflow.

There is a clear tendency for the simulated spectra to fall below the measured one at higher frequencies, in particular for the
outboard stations, which might be due to the resolution in the turbulence box which is 1.28 m in vertical and horizontal direction. Finally, it can be seen that in this case the difference between the two BEM implementations is quite small. This can be explained by the above considerations in Sect. 4 that if the local thrust coefficient is high the slope of the $u_{i,y}(U_0)$ curve in the grid BEM becomes almost horizontal and thus equal to the annular mean BEM. However, a light tendency of the annular mean BEM to overestimate the 1P and 2P on the two inboard stations with a lower thrust coefficient confirms the expected trend.





## 5.4 Yawed flow

In the New MEXICO experiments (Boorsma and Schepers, 2018), the aerodynamic loading on a 4.5 m diameter model turbine in uniform inflow and yawed inflow was measured. These measurements have been compared to results from many aerodynamic codes of different fidelities in (Schepers et al., 2018b). For a specific evaluation of the yaw modeling, Sect. 3.5, we look at the differences between the aerodynamic forces between the uniform and 30° yawed flow cases at roughly 15 m/s wind tunnel speed. In both cases the turbine had a rotor speed of 425.1 rpm, the blades were pitched at -2.3° and the tunnel speed was very similar at 15.06 m/s (axial flow) and 15.01 m/s (yawed flow). As such these measurements provide an excellent opportunity to validate the effect of yaw on both the mean and azimuthally varying load levels. Figure 25 shows the differences in normal (perpendicular to local chord) and tangential (parallel to local chord) forces at three measured sections at 25 %, 60 % and 82 % radius. These differences are computed as $\Delta F_{n/t} = F_{n/t,yaw} - F_{n/t,axial}$. Such a comparison involving both axial and yawed flow measurements and computations together was not included in (Schepers et al., 2018b).

It can be seen that there is a phase shift in the azimuthal force variation between the normal forces at the inboard section (top left plot of Fig. 25) and the section further outboard (bottom left plot). This phase shift is due to the dominance of either the root vortex (at the inboard section) or the tip vortex (outboard section). The root vortex is not accounted for in the present model and thus the HAWC2 computations do not agree well with the measured normal force at the inboard section. A recent engineering model that is based on high fidelity simulations and includes a correction for the root vortex is described by Rahimi et al. (2018).

For the sections further outboard, the influence of the tip vortex becomes more important and the phases of the azimuthal force variation agree well. There is a slight overprediction of the mean loading, especially in the tangential direction. Comparing the integrated out-of-plane and in-plane blade root bending moments in Fig. 26 shows that the phase difference seen in the inboard loads is not significant for the blade root moments. HAWC2 predicts a smaller reduction of the mean out-of-plane and in-plane moments, but the phases compare well to the measurements.





**Figure 25.** Comparison of the differences in the azimuthal distribution of normal forces (left plots) and tangential forces (right plots) predicted by HAWC2 with the forces measured in the New MEXICO experiment, at three different radial positions.



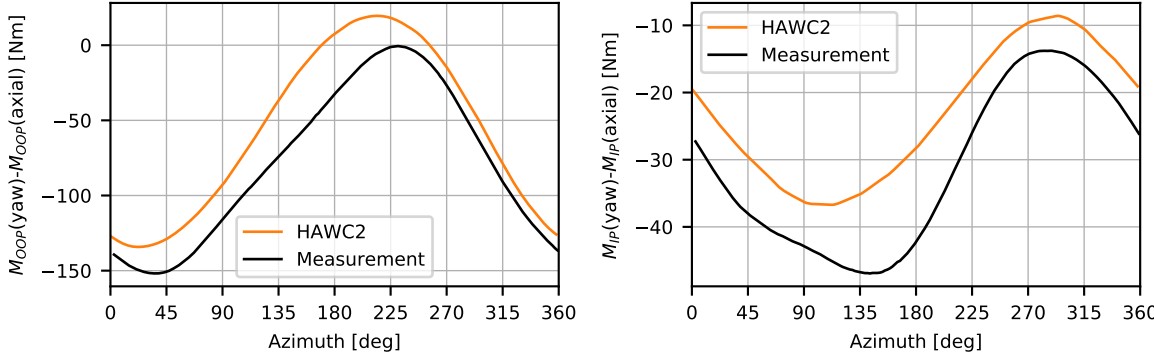

**Figure 26.** Comparison of differences in out-of-plane (left plot) and in-plane blade root bending moments (right plot) predicted by HAWC2 with the moments integrated from the measured forces in the New MEXICO experiment.

## 5.5 Dynamic inflow

The NREL/NASA Ames Phase VI experiments (Hand et al., 2001), performed in the NASA Ames open loop wind tunnel, include runs targeting dynamic inflow effects at 5.1 m s$^{-1}$ wind tunnel speed (denoted case *Q0500000*). The rotor speed was constant at 71.62 rpm and the pitch was varied 20 times at a rate of roughly 66° per second between -5.9° pitch (heavily loaded rotor, induction factor $a \approx 0.5$) and 10.02° pitch (unloaded rotor, $a \approx 0$). Averaging the responses of these 20 pitch steps shows pronounced dynamic inflow effects at all the instrumented blade sections at 30 %, 47 %, 63 %, 80 % and 95 % radius.

The measured data have been analysed by Schepers (2007) using a BEM code and a free wake code and by Sørensen and Madsen (2006) using a BEM code, a computational fluid dynamics code and a near wake model. More recently, this case was also used for comparison of various research codes in IEA Task 29, Phase 3 (Schepers et al., 2018b). An investigation of the radial dependency of the time constants in the force response, which seemed to reverse when the pitching direction was reversed, was conducted by Pirrung and Madsen (2018).

A comparison of measurements with the dynamic inflow model described in Sect. 3.6 is shown in Fig. 27. All the forces are scaled such that the pitch steps are between zero and one. This approach makes it possible to compare the dynamic response at different sections on the blade easily and was also used by (Schepers et al., 2018b). The computations assume a stiff turbine. It can be seen that the force overshoots at the 30 % section are generally larger than at the 80 % section. This is due to the slower response inboard due to the larger distance from the tip vortex. The dynamic inflow model takes this radial dependency of the time constants into account and the predicted overshoots of the forces are generally in good agreement with the measurements. An exception is that the overshoot of the tangential force at the inboard section (solid lines in the top right plot) is underestimated by HAWC2. The behavior of the tangential force for the pitching down case (bottom right plot) can be explained by a zero-crossing of the angle of attack at roughly 0.4 seconds. For both positive and negative AOA close to zero degrees the lift force has a component that is pointing towards the leading edge. Therefore the forces when pitching to




lower loading are decreasing until the AOA reaches roughly zero. Then the tangential forces increase as the AOA undershoots and then decrease again as the induced velocities decrease (causing AOA to increase and move closer to zero) towards the
equilibrium state at low loading.

The comparison shows good agreement, however some disagreements are to be expected due to inherent limitations of the actuator disc based model. Specifically, the root vortex dynamics are missing and the disc model also assumes an infinite number of blades. Therefore differences are expected close to the root and the tip of the blade, where the induction from a helical wake deviates most from the induction due to a cylindrical wake. An option to address these limitations is to couple a
vortex-based near wake model to the BEM code (Pirrung et al., 2016, 2017). However, the work in IEA Task 29 has shown that care has to be taken when coupling the induction dynamics (Schepers et al., 2018b).

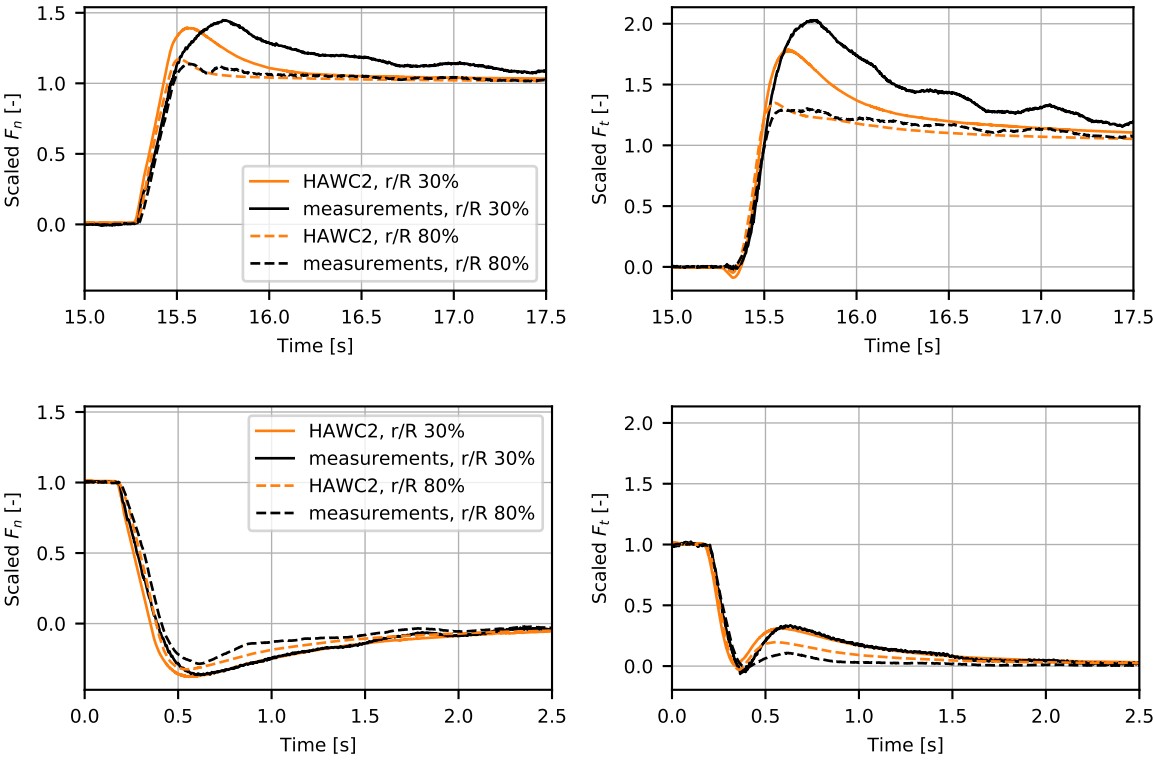

**Figure 27.** Comparison of HAWC2 results against measurements of the dynamic inflow case Q0500000 of the NREL/NASA Ames Phase VI experiment. The plots show scaled normal (left) and tangential (right) forces for pitch steps towards high loading (top) and low loading (bottom).





# 6 Conclusions

We have presented an implementation of the blade element momentum (BEM) method on a polar grid in order to simulate more accurately the considerable inflow and load variations over the rotor disc found for large turbines. The model can also
be characterised as an engineering actuator disc model where the induced velocities on the stationary polar grid are updated at each time step in an aeroelastic simulation. Further, the detailed integration of sub-models for tip correction, yaw and dynamic inflow has been described. Also a sub-model for radial induction important for computations with out-of-plane blades due to elastic effects or coning has been presented.

The load impact mechanism on the flapwise blade root moment from this unsteady induction by the grid BEM is analyzed. It
is found that the load impact strongly depends on the turbine design and operating conditions. For operation at low to medium thrust coefficients (conventional turbines at above rated wind speed or low induction turbines in the whole operating range) it is found that the grid BEM gives typically 8-10 % lower 1 Hz fatigue loads than the classical annular mean BEM approach. At high thrust coefficients the grid BEM can give slightly increased fatigue loads, in particular for pure shear cases

Different validation cases have been presented by comparing with experimental data and data from the high fidelity EllipSys3D
code. A challenging half wake in the vertical plane with the double inflow velocity on the one side of the rotor relative to the other side is simulated. A good correlation is found with EllipSys3D results for blade loads as function of azimuth.

Results on yawed inflow for the MEXICO rotor and dynamic inflow results from the NREL NASA Ames experiment confirm a satisfactory performance of the sub-models for yawed flow conditions and dynamic inflow. Finally, comparing PSD spectra of the simulated local aerodynamic forces at four radial positions on the full-scale NM80 turbine shows excellent agreement
with spectra of measured forces originating from the DanAero experiment.

*Author contributions.* HAaM and TJL developed and implemented the overall grid BEM modeling approach. TJL tested the grid BEM model and increased the robustness of the implementation with contributions from GRP. HAaM investigated the load mechanism of the grid BEM method. HAaM performed the actuator disc simulations and extracted the data for tuning the yaw and dynamic inflow model. HAaM, TJL and GRP wrote the article with contributions from FZ and AL. AL determined the time constants of the dynamic inflow model by means
of numerical optimization. TJL and AL derived and implemented the correction for blade in-plane and out-of-plane bending. GRP executed and discussed the validation cases with major contributions from HAaM. FZ derived the EllipSys3D set-up for the half wake simulations, conducted the simulations and extracted the data for the validation. All authors jointly finalized the paper.

*Competing interests.* The is no competing interest in the present paper.

*Acknowledgements.* We thank our colleagues in the AER and LAC section that in one or another way have contributed to this work and
the modeling presented. In particular the set-up for automatic pre- and post-processing the DLC1.2 simulations presented in section 4.5



developed by David Robert Verelst and Mads M Pedersen. Also the valuable contribution from Anders Melchior Hansen, one of the main developers of HAWC2, is acknowledged.

We also acknowledge the access to the New Mexico the NREL Nasa Ames data in the IEA Task 29 data base.





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
