# Peer review of "Implementation of the Blade Element Momentum Model on a Polar Grid and its Aeroelastic Load Impact"

_Wind Energy Science, 2019_

## Referee Comment (RC1) · Anonymous Referee #1 · 10 Sep 2019

Excellent paper, so I have only minor remarks: - Abstract; since it is an abstract (i.e. not an introduction) also the main results and conclusions should be mentioned

Greek letters: omega is missing (and perhaps also others)

line 97: this should be reformulated; In order for a small rotor to have an 1P the turbulent vortices should also be smaller than the rotor.

line 113: The subject of sheared and turbulent inflow was ....

Figure 5 (right): the differences are not clearly visible; perhaps the % differences can be mentioned

[Figure]

line 361, "Soerensen and Munduate" is missing in the references

line 442: mention the type of desktop PC which has been used

Section 4: include a table with the main properties of the AVATAR rotor (diameter, V_rated, Omega_rated)

Algorithm 1: Include a remark if skipping the azimuthal loops will lead to an annular mean BEM (in case not, mention what else have to be changed)

Figure 15 Mention if the 1P, 2P, etc. are also so clear for the other points (as indicated in Fig. 14).

Figure 16: for clarity, the same y-scaling should be used for the Left and Right graph

Figure 24: the unit of the PSD should be divided by Hz

References: in case of more references of the same author, they should be ordered to year References: Stepniewski is not at the correct order

Typo's: line 197: this cases Eq (3) + (4): skip "A" (denominator) Eq (5): skip "dr" Eq (6): be consistent with symbols: V_r (U_rel) and Omega (omega) Figure 18 + 19: annular mean BEM (instead of annual)

---

## Short Comment (SC1) · 2 Oct 2019

Dear authors,

It seems to me there might be a typo in equations 17 to 19, unless I misunderstood something. Considering a zero degree yaw angle, ka,i coefficients all vanish to zero (eq. 19). Thus, from equation (18), the ka coefficient is also zero (by the way, there is a double closing bracket here). We finally obtain zero induction, from equation (17). I did not checked, but in equation (17), ka should probably be replaced by (1-ka)?

Best regards.

---

## Short Comment (SC2) · 2 Oct 2019

Dear Frédéric,

Thank you very much for your comment!

There is indeed a typo in Equation 18, which should read:

$$k_a(C_{t,mean}) = k_{a,3}\min(C_{t,mean}, 0.9)^3 + k_{a,2}\min(C_{t,mean}, 0.9)^2 + k_{a,1}\min(C_{t,mean}, 0.9) + 1$$

Great that you found it so we can fix it in the revised version.

Best regards, Georg

---

## Referee Comment (RC2) · Anonymous Referee #2 · 6 Oct 2019

Very nice paper that justly stresses the importance of BEM implementations. Some comments and suggestions to further improve the paper:

-The paper points out differences between 'conventional' BEM (solving the equations for a whole annulus) and the grid based BEM implementation. Several of the aeroelastic codes mentioned in the intro make use of a local BEM approach, where the BEM equations are solved separately per blade using local flow conditions as input. Can the authors point out the difference of their implementation with respect to this more current BEM implementation, e.g. what differences can we expect in sheared and turbulent inflow?

-Section 3.4 page 10 The explanation on the grid based BEM is based on a 3-bladed rotor. What would be the approach for a 2-bladed rotor?

-Section 3.9 and Algorithm 1 The different time scales between dynamic stall and dynamic inflow effects are considered to justify the separate calculation of these effects rather than solve them in an integral fashion. Perhaps it is good to clarify whether the modification in lift due to dynamic stall effects still has an influence on the calculated induction and if so how? From the algorithm it appears the Beddoes Leishman dynamic arfoil data is calculated after the induction calculation, is that corrrect? And similarly, are the dynamic inflow and yaw correction applied in the BEM iterative loop for convergence or applied separately after that?

-Figure 16. To improve readability it is suggested to indicate in the figures (e.g. by adding a title, although it is indicated in the caption) what the difference is between the 2 plots?

-Section 4.3 page 25 line 500 It is mentioned that when operating in high thrust coefficients and non-uniform inflow conditions (e.g. shear), the local induced velocity can increase when the local wind speed decreases due to the high thrust coefficient and the corresponding slope in the Ct-a curve. As a consequence the fatigue seems to increase when the described local grid based BEM approach is used. Is it expected that this will physically occur as well (i.e. an increasing induced velocity for a decreasing inflow speed)? Has this effect been validated/verified against higher fidelity codes (e.g. CFD/vortex)?

---

## Author Comment (AC1) · 18 Oct 2019

Please find enclosed file with response to reviewer 1 and 2 and the updated paper with marked changes.

Please also note the supplement to this comment: https://www.wind-energ-sci-discuss.net/wes-2019-53/wes-2019-53-AC1-supplement.pdf

---

## Author Response (AR1)

October 18, 2019

hama

Dear Reviewers

First of all we would like to thank you for the constructive and detailed comments on our article. Please find below our responses (in black) to your comments (in blue). In addition to changes suggested by the reviewers we have added a few minor details on the code implementation. Please find at the end of this letter a marked-up version showing all changes in the paper.

Yours sincerely

HAa. Madsen, TJ. Larsen, GR. Pirrung, A. Li, and F. Zahle

Technical University of Denmark       Frederiksborgvej 399       hama@dtu.dk
**Department of Wind Energy**         Building 125               www.dtu.dk
                                      DK-4000 Roskilde

[Figure]

**Comments to Reviewer #1**

Excellent paper, so I have only minor remarks: - Abstract; since it is an abstract (i.e. not an introduction) also the main results and conclusions should be mentioned

We extended the abstract by the following sentence: *Comparisons with blade loads from full rotor CFD, wind tunnel experiments and a field experiment show that the model can predict the aerodynamic forces in half wake, yawed flow, dynamic inflow and turbulent inflow conditions.* However, we think we have already have included in the abstract the most important findings in the study:
*The load impact mechanism from this unsteady induction is analyzed and it is found that the load impact strongly depends on the turbine design and operating conditions. For operation at low to medium thrust coefficients (conventional turbines at above rated wind speed or low induction turbines in the whole operating range), it is found that the grid BEM gives typically 8-10 % lower 1 Hz fatigue loads than the classical annular mean BEM approach. At high thrust coefficients that can occur at low wind speeds the grid BEM can give slightly increased fatigue loads.*

Greek letters: omega is missing (and perhaps also others)

We have added $\alpha$, $\tau$ and $\omega$.

line 97: this should be reformulated; In order for a small rotor to have an 1P the turbulent vortices should also be smaller than the rotor.

We have reformulated the sentence to:
*In other words, for the increasing size of turbines a bigger and bigger part of the turbulent eddies have a size comparable to or below the rotor diameter.*

line 113: The subject of sheared and turbulent inflow was ....

Changed accordingly.

Figure 5 (right): the differences are not clearly visible; perhaps the % differences can be mentioned

We added the following sentences to the caption: *The error of the polynomial fit in the right plot is smaller than 3% for all shown yaw angles up to a CT of 0.87. For higher CT the deviations increase, especially for yaw angles above 45 degrees.*

line 361, "Soerensen and Munduate" is missing in the references

We have corrected an error in the referencing, correctly it is Sørensen, Shen and Munduate.

line 442: mention the type of desktop PC which has been used

We have added *on a 2016 workstation laptop*.

Section 4: include a table with the main properties of the AVATAR rotor (diameter, $V_{rated}$, $\Omega_{rated}$

We have included a table with turbine properties.

Algorithm 1: Include a remark if skipping the azimuthal loops will lead to an annular mean BEM (in case not, mention what else have to be changed)

First we have to mention that the loops for radial and azimuthal position were mixed up in the original manuscript. The loop for azimuthal position is the outer loop and for the radial position it's the inner one. This has now been corrected in the Algorithm 1 in the paper. This sequence of the azimuthal and radial loops complicates the conversion of the code to a mean annular version a little bit. What we did in this test version of HAWC2 was simply to run through the algorithm two times for each time step. In the first time step we stored the three wind speed components at each grid point in a new variable. At the end of the first loop we computed mean values over a ring element (constant r) of the three wind speed components. These wind speed components were then used in the second execution of the loop instead of the actual wind speed components at the grid points. We added the following text in the manuscript:

*Because the version is only a test version the mean annular approach was implemented in a crude way by executing the loop two times. During the first loop the local three wind speed components were summed in new variables for each grid point. At the end of the first loop the mean of the velocity components for a constants radius (a ring element) was derived and then used in the second loop instead of the local wind speed components.*

Figure 15 Mention if the 1P, 2P, etc. are also so clear for the other points (as indicated in Fig. 14).

We added the following to the caption of Figure 15: *Clearly the induced velocity at the blade exhibits 1P, 2P, . . ., nP peaks corresponding to the rotationally sampled turbulent inflow. These peaks can be observed on other radial positions on the blade as well.*

Figure 16: for clarity, the same y-scaling should be used for the Left and Right graph

Changed accordingly.

Figure 24: the unit of the PSD should be divided by Hz

Changed accordingly.

References: in case of more references of the same author, they should be ordered to year
References: Stepniewski is not at the correct order

We agree. We used the LaTeX packages provided by WES. We think this will be fixed at the typesetting stage.

Typo's: line 197: this cases Eq (3) + (4): skip "A" (denominator) Eq (5): skip "dr" Eq (6): be consistent with symbols: $V_r$ ($U_{rel}$) and Omega (omega) Figure 18 + 19: annular mean BEM (instead of annual) Changed accordingly.

[Figure]

**Comments to Reviewer #2**

Very nice paper that justly stresses the importance of BEM implementations. Some comments and suggestions to further improve the paper: -The paper points out differences between 'conventional' BEM (solving the equations for a whole annulus) and the grid based BEM implementation. Several of the aeroelastic codes mentioned in the intro make use of a local BEM approach, where the BEM equations are solved separately per blade using local flow conditions as input. Can the authors point out the difference of their implementation with respect to this more current BEM implementation, e.g. what differences can we expect in sheared and turbulent inflow?

We are not aware that our implementation of the BEM model as an engineering actuator disc (AD) on a polar grid has been used or presented by other researchers. With the present implementation we follow the basic idea behind the actuator disc concept by distributing the loading over the complete disc and in this case a non-uniform loading which is updated at each time step. The dynamic inflow filter can then be applied on this disc flow. Finally, the rotating blades scan through this disc induced flow field and the mechanism can be characterized as a rotational sampling of the induced flow field.

We find that this is fundamentally different from e.g. using a sector implementation of the BEM model which makes it difficulty to implement a dynamic inflow model without damping the 1p, 2p etc. variations of the induced flow field or adding a phase shift. A damping and phase shift of the 1p, 2p, etc. of the induced velocities will in general lead to higher fatigue loads.

An important mechanism of the induction of the presented BEM implementation on a polar grid is that each grid point has a memory effect incorporated. Thus past loading changes at a grid point (e.g. due to a pitch action in this region, a local gust, an instantaneous shear, a blade passing with another pitch angle offset) will influence the induction of the blade passing that grid point. The weighting of the impact of these past events is controlled by the dynamic inflow filter.

To clarify even better the characteristics of the present implementation we added the above last paragraph into the section 4.1 of the paper:

*An important mechanism of the induction of the presented BEM implementation on a polar grid is that each grid point has a memory effect incorporated. Thus past loading changes at a grid point (e.g. due to a pitch action in this region, a local gust, an instantaneous shear, a blade passing with another pitch angle offset) will influence the induction of the blade passing that grid point. The weighting of the impact of these past events is controlled by the dynamic inflow filter.*

-Section 3.4 page 10 The explanation on the grid based BEM is based on a 3-bladed rotor. What would be the approach for a 2-bladed rotor?

The approach would be the same by just changing the parameter $N_B$ in the equations. The induced velocities are still computed based on local inflow conditions at the grid points. The only increased uncertainty is on the interpolation of the two blade sections velocity, deflection, pitch angle and twist to the grid points which now can be at a maximum distance of 90 deg of the two neighbouring blades for a two-bladed rotor instead of 60 deg for a three-bladed rotor.

-Section 3.9 and Algorithm 1 The different time scales between dynamic stall and dynamic inflow effects are considered to justify the separate calculation of these effects rather than solve them in an integral fashion. Perhaps it is good to clarify whether the modification in lift due to dynamic stall

[Figure]

*effects still has an influence on the calculated induction and if so how? From the algorithm it appears the Beddoes Leishman dynamic arfoil data is calculated after the induction calculation, is that corrrect? And similarly, are the dynamic inflow and yaw correction applied in the BEM iterative loop for convergence or applied separately after that?*

It is correct that the dynamic airfoil data is computed after the induction computation. Thus the dynamic stall model has no influence on the calculated induction. The dynamic stall effect is very fast compared to the dynamic inflow with time constants proportional to half chord / relative speed compared to diameter / wind speed. The diameter is 1-2 orders of magnitude larger than the half chord and the relative speed is typically much higher than the wind speed. But it is true that this might lead to a small azimuth shift of the induced velocity field if there is consistent dynamic stall happening at the same azimuth angle with each rotation, for example in extreme yaw or shear. There is no BEM convergence loop. The quasi-steady induced velocities are instead input to the dynamic inflow filter. The induced velocities then converge in time, rather than in a single time step. The yaw correction is applied before the dynamic inflow filter, see Section 3.9.

*-Figure 16. To improve readability it is suggested to indicate in the figures (e.g. by adding a title, although it is indicated in the caption) what the difference is between the 2 plots?*

Changed accordingly.

*-Section 4.3 page 25 line 500 It is mentioned that when operating in high thrust coefficients and non-uniform inflow conditions (e.g. shear), the local induced velocity can increase when the local wind speed decreases due to the high thrust coefficient and the corresponding slope in the Ct-a curve. As a consequence the fatigue seems to increase when the described local grid based BEM approach is used. Is it expected that this will physically occur as well (i.e. an increasing induced velocity for a decreasing inflow speed)? Has this effect been validated/verified against higher fidelity codes (e.g. CFD/vortex)?*

It's clear that at high thrust coefficients above e.g. 0.89 the BEM momentum equations are not valid and the empirical relation $C_T$ vs. $a$ based on high fidelity AD simulations or experiments must be used. In the referenced study Madsen (1997) for a uniformly loaded AD at high thrust coefficients above 1 it is found that the induction starts to be a function of radial position and also increasing fast as function of increasing $C_T$. So if an increasing induced velocity is possible for a decreasing local free stream velocity and thus strongly increasing $C_T$ for a highly loaded rotor we have not proved. However, the HAWC2 code with the BEM implementation has now been coupled to the MIRAS Vortex code:
*Ramos García, Néstor, et al. "Validation of a Three-Dimensional Viscous-Inviscid Interactive Solver for Wind Turbine Rotors." Renewable Energy, vol. 70, Elsevier Ltd, 2014, pp. 78–92, doi:10.1016/j.renene.2014.04.001.*
which will enable to study these details.
Finally, it should be noted that for normal inflow conditions with a combination of shear and turbulence (DLC1.2) the grid BEM gives, even for a highly loaded rotor as the DTU 10MW turbine, considerable lower fatigue loads than the annular mean BEM for most wind speeds.

[revised manuscript text omitted]